



# Validation of Aeolus Level 2B wind products using wind profilers, ground-based Doppler wind lidars, and radiosondes in Japan

Hironori Iwai[1], Makoto Aoki[1], Mitsuru Oshiro[1], and Shoken Ishii[2]

[1]Radio Research Institute, National Institute of Information and Communications Technology, 4-2-1 Nukuikita, Koganei,
Tokyo 184-8795, Japan
[2]Department of Aeronautics and Astronautics, Tokyo Metropolitan University, 6-6 Asahigaoka, Hino, Tokyo 191-0065,
Japan

*Correspondence to*: Hironori Iwai (iwai@nict.go.jp)

**Abstract.** The first space-based Doppler wind lidar (DWL) onboard the Aeolus satellite was launched by the European
Space Agency (ESA) on 22 August 2018 to obtain global profiles of horizontal line-of-sight (HLOS) wind speed. In this
study, the Raleigh-clear and Mie-cloudy winds for periods of baseline 2B02 (from 1 October to 18 December 2018) and
2B10 (from 28 June to 31 December 2019 and from 20 April to 8 October 2020) were validated using 33 wind profilers
(WPRs) installed all over Japan, two ground-based coherent Doppler wind lidars (CDWLs), and 18 GPS-radiosondes (GPS-
RSs). In particular, vertical and seasonal analyses were performed and discussed using WPR data. During the baseline 2B02
period, a positive bias was found to be in the ranges of 0.46–1.69 m s$^{-1}$ for Rayleigh-clear winds and 1.63–2.42 m s$^{-1}$ for
Mie-cloudy winds using the three independent reference instruments. The biases of Rayleigh-clear and Mie-cloudy winds
were in the ranges of –0.82–+0.45 m s$^{-1}$ and –0.71–+0.16 m s$^{-1}$ during the baseline 2B10 period, respectively. The
systematic error for the baseline 2B10 was improved as compared with that for the baseline 2B02. The vertical analysis
using WPR data showed that the systematic error was slightly positive in all altitude ranges up to 11 km during the baseline
2B02 period. During the baseline 2B10 period, the systematic errors of Rayleigh-clear and Mie-cloudy winds were improved
in all altitude ranges up to 11 km as compared with the baseline 2B02. Immediately after the launch of Aeolus, both
Rayleigh-clear and Mie-cloudy biases were small. Within the baseline 2B02, the Rayleigh-clear and Mie-cloudy biases
showed a positive trend. For the baseline 2B10, the Rayleigh-clear wind bias was generally negative at all months except
August 2020, and Mie-cloudy wind bias gradually fluctuated. The systematic error was close to zero with time in 2020 and
did not show a marked seasonal trend. The dependence of the Rayleigh-clear wind bias on the scattering ratio was
investigated, showing that the scattering ratio had a minimal effect on the systematic error of the Rayleigh-clear winds
during the baseline 2B02 period. On the other hand, during the baseline 2B10 period, there was no significant bias
dependence on the scattering ratio. Without the estimated representativeness error associated with the comparisons using
WPR observations, the Aeolus random error was determined to be 6.71 (5.12) and 6.42 (4.80) m s$^{-1}$ for Rayleigh-clear (Mie-
cloudy) winds during the baseline 2B02 and 2B10 periods, respectively. The main reason for the large random errors is
probably related to the large representativeness error due to the large sampling volume of the WPRs. Using the CDWLs, the
Aeolus random error estimates were in the range of 4.49–5.31 (2.93–3.19) and 4.81–5.21 (3.30–3.37) m s$^{-1}$ for Rayleigh-





clear (Mie-cloudy) winds during the baseline 2B02 and 2B10 periods, respectively. By taking the GPS-RS representativeness error into account, the Aeolus random error was determined to be 4.01 (3.24) and 3.02 (2.89) m s$^{-1}$ for
Rayleigh-clear (Mie-cloudy) winds during the baseline 2B02 and 2B10 periods, respectively.

## 1 Introduction

Understanding the evolution and structure of winds is very important for numerical weather prediction (NWP). Wind is one of the fundamental meteorological variables describing the atmospheric state. Measurement of the three-dimensional global
wind field is crucial for NWP, air quality monitoring and forecasting, climate studies, and various meteorological studies. Accurate NWP is useful for commercial activities such as agriculture, fisheries, construction, transportation, and energy development, and for daily life. The current global observation network contains various wind measurements such as radiosondes, wind profilers (WPRs), ground-based Doppler wind lidars (DWLs), and aircrafts. The current global observation network provides accurate and precise vertical wind profiles. However, the observational coverage is limited
from the global perspective. Wind vectors can be measured by satellite-borne microwave scatterometers and polarimetric microwave radiometers, and the multiple-layer wind vector, called the atmospheric motion vector (AMV), can be retrieved from cloud and water vapour motions derived from geostationary and polar-orbit satellite images. Although these sensors have a large coverage area and high temporal and horizontal resolutions, they lack or have significantly limited vertical sounding capability.
A space-based DWL is a powerful remote sensing instrument for global wind profiling. The European Space Agency (ESA) launched on 22 August 2018 the first space-based DWL, Aeolus, for obtaining global wind profiles (Kanitz et al., 2019; Reitebuch et al., 2020). Aeolus carries a single payload, named Atmospheric Laser Doppler Instrument (ALADIN). ALADIN uses a single-frequency UV laser and a direct-detection system and provides profiles of a single line-of-sight (LOS) wind speed on a global scale from the ground up to about 30 km in the stratosphere (ESA, 1999; Stoffelen et al., 2005,
2020; Reitebuch, 2012; Kanitz et al., 2019). The main purpose of Aeolus is to provide global wind profiles with vertical resolution and wind observation accuracy that meet for the World Meteorological Organization (WMO) observation requirements to improve NWP and to fill the gap of the current global wind observation systems. Its other main purpose is to contribute to research on the energy balance, atmospheric circulation, precipitation system, southern vibration phenomenon, stratosphere/troposphere exchange, and so forth.
The new remote sensing technology and retrieval algorithm require a careful assessment of the quality and validity of the generated data products before releasing them to the user community. ESA released an Announcement of Opportunity (AO) in 2007 and 2014 calling for calibration and validation (CAL/VAL) proposals for Aeolus. The CAL/VAL activities include a full assessment of all aspects of the DWL wind measurement performance and stability. The National Institute of Information and Communications Technology (NICT) has applied to contribute to CAL/VAL activities for Aeolus in East



Asia and the Western Pacific region. Continuous validation of horizontal LOS (HLOS) wind speed after calibration processes is important in order to contribute to the L2C product, which results from the background assimilation of the Aeolus HLOS winds in the European Centre for Medium-Range Weather Forecasts (ECMWF) operational prediction model. The purposes of the project are to contribute to reducing uncertainty in Aeolus wind measurements, to validate processes for improving HLOS wind speed measured by Aeolus, and to assess the quality of wind data.

The aim of this paper is therefore to validate the quality of the Aeolus HLOS winds over Japan by using measurements from WPRs, ground-based coherent Doppler wind lidars (CDWLs), and GPS-radiosondes (GPS-RSs). The paper is organized as follows. First, an overview of Aeolus and ALADIN is provided. Section 3 describes the WPR, CDWL, and GPS-RS instrument setups and measurement procedures. The procedure of matching the Aeolus measurements with the reference instruments' measurements is also described in Sect. 3. The intercomparison and statistical methods are addressed in Sect. 4.

Section 5 presents statistical comparisons between the Aeolus measurements and the WPR, CDWL, and GPS-RS measurements. In Sect. 6, the main findings are summarized.

## 2 Overview of Aeolus and ALADIN

Aeolus flies in a sun-synchronous polar orbit (inclination 97°) at an altitude of about 320 km, with a period of about 90 min

and a seven-day repeat cycle. The typical ground tracks of Aeolus over Japan are shown in Fig. 1. The red and blue lines

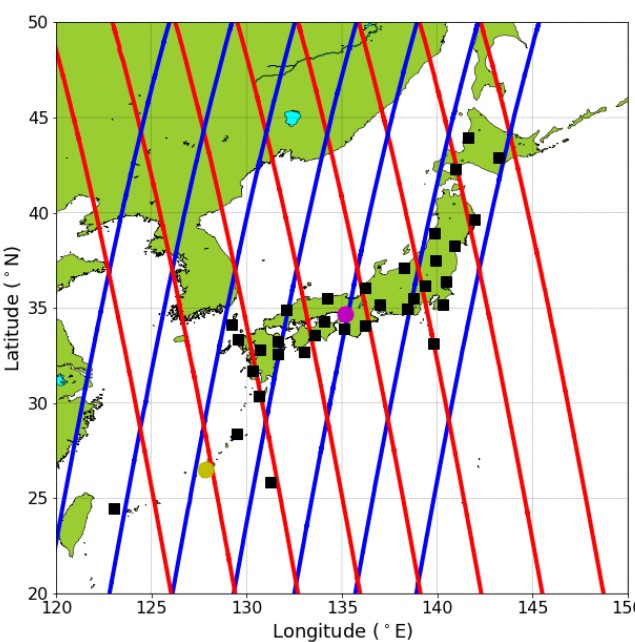

**Figure 1.** Map showing the locations of WPRs (black squares), Kobe CDWL (magenta circle), and Okinawa CDWL (yellow circle). Red and bule lines represent the typical Aeolus ground tracks for ascending and descending orbits, respectively.



represent the Aeolus ground tracks for ascending and descending orbits, respectively. The principal components of ALADIN are two fully redundant diode-pumped single-frequency continuous-wave neodymium-doped yttrium-aluminium-garnet (Nd:YAG) lasers and two diode-pumped Q-switched Nd:YAG lasers (Flight Model A (FM-A) and FM-B) with power amplifiers, a 1.5-m-diameter afocal Cassegrain telescope, a direct-detection receiver, and signal processing devices. The

single-frequency Q-switched Nd:YAG lasers with a 1064.4 nm operating wavelength emit about 250 mJ output energy with a 20 ns pulse width (full width at half maximum) operating at a pulse repetition frequency (PRF) of 50.5 Hz. Nonlinear lithium triborate crystals are used to generate the UV laser pulses with a 354.8 nm operating wavelength. The single-frequency Q-switched UV laser emits about 60 mJ output energy at the PRF of 50.5 Hz (Lux et al., 2020a) and a laser beam divergence of 20 µrad. The laser pulses are directed downward to Earth at an off-nadir angle of 35° and enter at an incident

angle of about 37.6° at the sea and land surfaces due to Earth's curvature. The FM-A laser was used until the middle of June 2019, and the FM-B laser has been used since 28 June 2019. The direct-detection receiver consists of the Cassegrain telescope, two interferometers, and two accumulation charge-coupled devices (ACCDs). The signal backscattered by moving atmospheric molecules (Rayleigh scattering) and aerosol and cloud particles (Mie scattering) is collected by the afocal Cassegrain telescope. One of the two interferometers uses the double-edge technique using two Fabry–Perot

interferometers (Chanin et al., 1989; Flesia and Korb, 1999; Flesia and Hirt, 2000; Gentry et al., 2000), which is mainly sensitive to atmospheric molecules (Rayleigh channel). The other one uses a spectrometer based on a Fizeau interferometer (Schillinger et al., 2003; Morancais et al., 2004), which is sensitive to aerosol and cloud particles (Mie channel). Both the Fizeau and Fabry–Perot interferometers act as a narrowband filter. The signals for Rayleigh and Mie channels are imaged on each ACCD after passing through some optics (Weiler et al., 2020). The signals imaged on the two ACCDs are converted to

electrical signals and stored.

We used the Aeolus Level 2B (L2B) data products of Raleigh and Mie channels during the baseline 2B02 and 2B10 periods. The L2B data products including a primary mirror correction (baseline 2B10; Rennie and Isaksen, 2020) have been available for new observations since April 2020. The homogeneous reprocessed data are also available using the baseline 2B10 from 28 June to 31 December 2019. In this study, we used three different periods to assess L2B data products, from 1

October to 18 December 2018 (baseline 2B02), and from 28 June to 31 December 2019 and from 20 April to 8 October 2020 (baseline 2B10), where a consistent re-analysed dataset is available. The first period was within the commissioning phase. We mainly discuss the measurement performance of Aeolus for Rayleigh-clear and Mie-cloudy winds. Rayleigh-clear winds refer to wind observations in an aerosol-free atmosphere. Mie-cloudy winds refer to winds acquired from Mie backscattered signals induced by aerosols and clouds (Witschas et al., 2020). The quality of the Aeolus wind data is indicated by validity

flags. The validity flag (de Kloe et al., 2016) considers the validity of the products. Several different technical, instrumental, and retrieving checks account for this flag. It has the value 1 (valid) or 0 (not valid). We only used Aeolus products with a validity flag of 1. We also used HLOS estimated errors (theoretical) of the L2B data products. The estimated error is a theoretical value, which is estimated on the basis of measured signal levels as well as the temperature and pressure sensitivities of the Rayleigh channel response (Dabas et al., 2008).






## 3 Overview of reference instruments

### 3.1 Wind profilers

In April 2001, the Japan Meteorological Agency (JMA) started the operation of a wind profiler (WPR) network, WInd profiler Network and Data Acquisition System (WINDAS; Ishihara et al., 2006). WINDAS consists of 33 1.3-GHz-band wind profilers as of August 2021 (black squares in Fig. 1). The specifications of WPR are listed in Table 1. WINDAS can operate continuously, acquiring vertical profiles of horizontal wind speed, wind direction, vertical velocity, and signal-to-noise ratio (SNR) over the wind profilers using five beams (one vertical beam and four oblique beams). The horizontal wind speed and wind direction are calculated from radial wind speeds by the four-beam method under strict data quality control (Adachi et al., 2005). WINDAS provides a profile of wind data with high accuracy. In operational mode, the temporal and vertical resolutions of WINDAS data are 10 min and 291 m, respectively. The minimum and maximum detection heights are 294m and 11.6 km above the wind profiler, respectively. There are 40 range bins for one wind profile. There is no significant difference between wind profiler winds and radiosonde winds in the biases and root mean square errors (Ishihara et al., 2006). The random error (root mean square error) $\sigma_{WPR}$ of zonal winds was determined to be about 3 m s$^{-1}$. The comparison

**Table 1.** Specifications of WPRs.

| Transmitter | |
|---|---|
| Frequency (GHz) | 1.35 |
| Peak power (kW) | 4.8 |
| Pulse repetition frequency (kHz) | 5, 10, 15, 20 10 (Operation) |
| Pulse width (µs) | 0.67, 1.33, 2, 2.66, 4 |
| Beam width (degree) | 3.9 |
| Beam elevation angle (degree) | 76, 90 |
| Beam azimuth angle (degree) | 0, 90, 180, 270 |
| Number of beams | 5 |
| **Receiver** | |
| Antenna | Active phased array antenna |
| Observation altitude range (m) | 294 – 11,600 |
| Range solution (m) | 100, 150, 200, 300, 400, 600 |
| Vertical resolution (m) | 291 (Operation) |
| Temporal resolution for wind measurement (min) | 1 |
| Temporal resolution for averaging (min) | 10 |





of wind data between Aeolus and the WPRs is useful for assessing wind measurement performance and the spatiotemporal
variation in the wind field.

Considering the different spatial and temporal resolutions of the WPRs and the Aeolus, data matching procedures are
necessary before comparing the data obtained by the two sensors. First, the WPR data and Aeolus data need to be matched in
both space and time. To achieve geographical matching, the distance between the mean positions of an Aeolus measurement
and the WPR is set to be less than 100 km. To achieve temporal synchronization, we use averages of WPR wind data from
30 min before to 30 min after the passage of Aeolus. After temporal and spatial collocation, the Aeolus L2B wind product
closest to each WPR measurement is adopted for comparison. The horizontal wind speed and wind direction measured by
the WPRs during the periods from 1 October 2018 to 15 May 2019 (baseline 2B02) and from 28 June to 31 December 2019
and from 20 April to 8 October 2020 (baseline 2B10) were used to compare Aeolus HLOS wind data.

**3.2 Coherent Doppler wind lidars**

NICT has installed 1.54-μm CDWLs (WINDCUBE 400S manufactured by LEOSPHERE; Cariou et al., 2006) in Kobe
(34.66ºN, 135.16ºE; magenta circle in Fig. 1) and Okinawa (26.50ºN, 127.84ºE; yellow circle in Fig. 1). The specifications
of the CDWLs are listed in Table 2. The CDWL in Kobe was placed on the rooftop of a building managed by Kobe City.
The CDWL in Okinawa was placed on the fifth floor (25.1 m MSL) of the steel tower in Okinawa Electromagnetic

**Table 2.** Specifications of CDWLs.

| Transmitter | |
|---|---|
| Wavelength (μm) / Frequency (THz) | 1.543 / 194 |
| Average power (W) | 1.8 |
| Pulse repetition frequency (kHz) | 10 |
| Pulse width (ns) | 800 |
| Laser beam elevation angle (degree) | -10 – +190 |
| Laser beam azimuth angle (degree) | 0 – 360 |
| Number of beams | 5 |
| Receiver | |
| Telescope diameter with 2-axis scanning device (m) | 0.12 |
| Observation altitude range (m) | 300 – 13,400 |
| Range resolution (m) | 150 |
| Temporal resolution for wind measurement (s) | 10 |
| Temporal resolution for averaging (min) | 60 |





Technology Center of NICT (hereafter, NICT Okinawa). In this experiment, their range bins had a length of 150 m with the center of the first bin at 300 m. With 159 range bins per beam, adjacent range bins were overlapped by 83.1 m and the maximum range was about 13.4 km depending on the aerosol load and/or cirrus clouds present. The vertical profiles of horizontal wind speed and wind direction were acquired by the Doppler beam swinging (DBS) technique from four inclined beams (north, east, south, and west) with an elevation angle of 70°. The Doppler velocity spectra for all range bins of each

beam were obtained 10,000 times on average. Since the PRF was 10 kHz, the accumulation time of each beam was 10 s. The Doppler wind speed at each bin was estimated from the averaged Doppler-shifted frequency spectra using the maximum likelihood estimator. We evaluated the bias and random error for wind measurements of the CDWLs using the methods described by Iwai et al. (2013). Bias was estimated at 0.02 m s⁻¹ using measurements from a stationary hard target. Random errors were 0.02 to 0.10 m s⁻¹ from –10 to –30 dB wideband SNR and the CDWLs operate near a theoretical Cramer–Rao

lower bound (Aoki et al., 2016; Rye and Hardesty 1993). On the basis of the comparison with collocated radiosonde data, the systematic error and random error (root mean square error) $\sigma_{CDWL}$ of horizontal wind speed acquired by the DBS technique were determined to be about 0.2 and 2 m s⁻¹, respectively (Aoki et al., 2015). Therefore, the CDWL measurements act as a reference owing to their low systematic and random errors that result from the coherent measurement principle of the system. As for the WPR data, the CDWL data and Aeolus data need to be matched in both space and time. To achieve geographical

matching, the distance between the mean position of an Aeolus measurement and the CDWL should be less than 100 km. As mentioned earlier, we averaged Doppler velocity spectra for all range bins of each beam from 30 min before to 30 min after the passage of Aeolus, and then the vertical profiles of horizontal wind speed and wind direction were acquired by the DBS technique. In Okinawa, the vertical profiles of horizontal wind speed and wind direction measured during the periods from 18 October 2018 to 11 May 2019 (baseline 2B02) and from 28 June to 31 December 2019 and from 20 April to 8 October

2020 (baseline 2B10) were obtained to compare Aeolus HLOS wind data. In Kobe, the vertical profiles of horizontal wind speed and wind direction measured during the periods from 16 October 2018 to 15 May 2019 (baseline 2B02) and from 3 September to 31 December 2019 and from 20 April to 15 July 2020 (baseline 2B10) were obtained to compare Aeolus HLOS wind data.

**3.3 Radiosondes**

Twelve GPS-radiosondes (GPS-RSs) of type RS41-SGP produced by Vaisala were launched from NICT Okinawa (26.50ºN, 127.84ºE; yellow circle in Fig. 1) from October to December 2018 (baseline 2B02). The specifications of the RS41-SGP are listed in Table 3. From September to December 2019 (baseline 2B10), six GPS-RSs were also launched from NICT Okinawa. An overview of the 18 obtained validation cases is given in Table 4. The GPS-RSs transmit observed data every 2 s to an

MW41 ground receiver unit. The observed data are processed using Vaisala proprietary software (DigiCORA version). The vertical resolution is about 10 m at the typical ascending speed of 5 m s⁻¹. The horizontal wind speed and direction are calculated using changes in the GPS location. According to the estimated Global Climate Observing System Reference





**Table 3.** Specifications of GPS-RSs of type RS41-SGP.

| | |
|---|---|
| Wind speed | |
|     Resolution (m s⁻¹) | 0.1 |
|     Velocity measurement uncertainty (m s⁻¹) | 0.7 |
|     Maximum reported wind speed (m s⁻¹) | 160 |
| Wind direction | |
|     Resolution (degree) | 0.1 |
|     Directional measurement uncertainty (degree) | 1 |
|     Wind direction range (degree) | $0 - 360$ |
| Geopotential height | |
|     Resolution (gpm) | 0.1 |
|     Measurement range (gpm) | Surface to 40,000 |
|     Accuracy (gpm) | 10.0 |

Upper-Air Network (GRUAN), the measurement uncertainties of the horizontal wind speed $\sigma_{GPS-RS}$ and direction are assumed to be 0.7 m s⁻¹ and 1°, respectively (Dirksen et al., 2014). Although the measurement uncertainties are derived from the radiosonde of type RS92 and not RS41, there is no significant difference in the uncertainty as both radiosonde types use the same technique to obtain wind speed and direction (Jensen et al., 2016; Kawai et al., 2017). Since the GPS-RS wind data are obtained by direct in situ measurements, the GPS-RS observations are generally very accurate and the instrument errors are small. The GPS-RS measurements are suitable for use as a reference data set for the validation of Aeolus HLOS winds. Furthermore, the observation errors can be assumed to be uncorrelated between different GPS-RSs. However, other errors arise due to the GPS-RS drift during its ascent. The averaged ascent time of the GPS-RSs is about 45 min when they reach an altitude of 25 km. The GPS-RSs launched from NICT Okinawa drifted by a horizontal distance of up to about 120 km. These values are considered when defining collocation criteria for comparisons of Aeolus and GPS-RS measurements. In this study, the GPS-RS measurements that are within 120 km horizontal distance and 60 min temporal difference from the Aeolus measurements are used for the validation.

**4 Intercomparison and statistical methods**

There is a difference in the vertical resolution between Aeolus measurements and WPR, CDWL, and GPS-RS measurements. The horizontal wind speed and wind direction measured by the WPRs, CDWLs, and GPS-RSs are averaged to the Aeolus bin by using the top and bottom altitudes given in the Aeolus L2B data product. All valid averaged wind speeds





**Table 4.** Overview of Aeolus validation cases obtained with GPS-RS launched at NICT Okinawa for baselines 2B02 and 2B10. The baseline, date, GPS-RS launch time, and Aeolus overpass time are given. The last column indicates whether Aeolus had an ascending or a descending orbit.

| Baseline | Date | GPS-RS launch time (UTC) | Aeolus overpass time (UTC) | Aeolus orbit type |
|---|---|---|---|---|
| 2B02 | 1 Nov 2018 | 21:21 | 21:35 | Descending |
| | 8 Nov 2018 | 21:20 | 21:35 | Descending |
| | 10 Nov 2018 | 09:08 | 09:22 | Ascending |
| | 15 Nov 2018 | 21:19 | 21:35 | Descending |
| | 24 Nov 2018 | 09:07 | 09:22 | Ascending |
| | 29 Nov 2018 | 21:20 | 21:34 | Descending |
| | 1 Dec 2018 | 09:07 | 09:22 | Ascending |
| | 6 Dec 2018 | 21:20 | 21:35 | Descending |
| | 8 Dec 2018 | 09:07 | 09:22 | Ascending |
| | 13 Dec 2018 | 21:20 | 21:34 | Descending |
| | 15 Dec 2018 | 09:07 | 09:21 | Ascending |
| | 20 Dec 2018 | 21:20 | 21:35 | Descending |
| 2B10 | 19 Sep 2019 | 22:06 | 21:35 | Descending |
| | 7 Nov 2019 | 21:20 | 21:35 | Descending |
| | 9 Nov 2019 | 09:07 | 09:22 | Ascending |
| | 23 Nov 2019 | 09:07 | 09:22 | Ascending |
| | 19 Dec 2019 | 21:20 | 21:34 | Descending |
| | 21 Dec 2019 | 09:07 | 09:22 | Ascending |

($ws_{i=WPR,CDWL,GPS-RS}$) and directions ($wd_{i=WPR,CDWL,GPS-RS}$) are projected onto the HLOS wind speed of Aeolus ($HLOS_{i=WPR,CDWL,GPS-RS}$) by means of the Aeolus azimuth angle $\varphi_{Aeolus}$ according to the following equation (Witschas et al., 2020):

$$HLOS_i = \cos(\varphi_{Aeolus} - wd_i) \cdot ws_i. \qquad (1)$$

To validate the quality of Aeolus HLOS winds ($HLOS_{Aeolus}$), the difference from the corresponding WPR, CDWL, and

GPS-RS winds projected onto the Aeolus viewing direction ($HLOS_{WPR/CDWL/GPS-RS}$) is calculated according to

$$HLOS_{diff} = HLOS_{Aeolus} - HLOS_{WPR/CDWL/GPS-RS}. \qquad (2)$$

Following Witschas et al. (2020), the difference between Aeolus HLOS winds and WPR HLOS winds ($HLOS_{diff}$) can be used to verify the thresholds for the estimated HLOS error provided in the Aeolus L2B data product during the baseline





2B02 and 2B10 periods as shown in Figs. 2 and 3, respectively. For the Rayleigh-clear winds (Figs. 2a and 3a), the lowest
estimated HLOS errors are 2.3 and 2.3 m s$^{-1}$ during the baseline 2B02 and 2B10 periods, respectively. The HLOS
differences remain reasonably constant until an estimated HLOS error of about 8 m s$^{-1}$ and then increases with increasing
estimated HLOS error. The Mie-cloudy winds (Figs. 2b and 3b) show estimated HLOS errors of as little as 0.18 and 0.4 m
s$^{-1}$ during the baseline 2B02 and 2B10 periods, respectively. The HLOS differences are reasonably constant up to an
estimated error of about 5 m s$^{-1}$ and then show a considerable increase for larger estimated HLOS errors. Therefore, only
Rayleigh-clear winds with estimated HLOS errors smaller than 8 m s$^{-1}$ and Mie-cloudy winds with estimated HLOS errors
smaller than 5 m s$^{-1}$ are used for the validation. These estimated HLOS error thresholds are consistent with
recommendations of the Aeolus CAL/VAL teams (Rennie and Isaksen, 2020) and those adopted in other validation studies
(e.g., Baars et al., 2020).

To evaluate the results of comparison between Aeolus HLOS winds and reference instruments' HLOS winds, we use mean
differences (BIAS) and the standard deviation (STD) of the differences as:

$$\text{BIAS} = \frac{1}{N}\sum_{i=1}^{N} HLOS_{diff}(i) \,, \tag{3}$$

$$\text{STD} = \sqrt{\frac{1}{N-1}\sum_{i=1}^{N}\big(HLOS_{diff}(i) - \text{BIAS}\big)^2} \,, \tag{4}$$

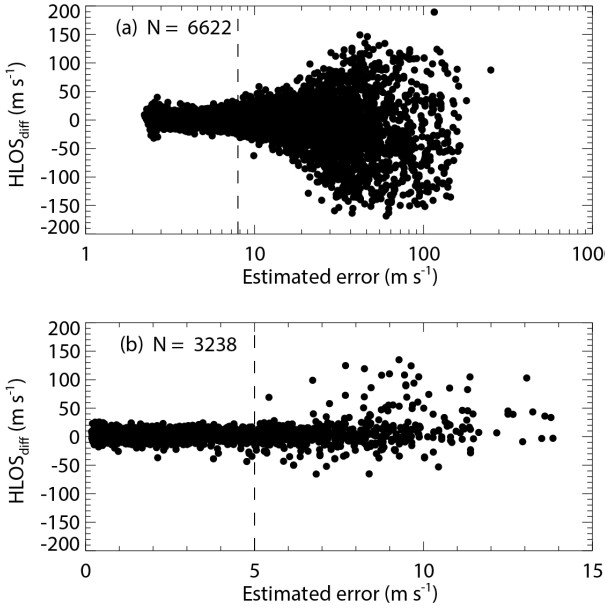

**Figure 2.** Dependence of wind speed difference between the Aeolus HLOS and WPR HLOS winds on the estimated HLOS error
given in the L2B product for (a) Rayleigh-clear winds and (b) Mie-cloudy winds for baseline 2B02. The areas on the right of the
vertical dashed lines indicate the data with estimated errors larger than 8 m s$^{-1}$ (Rayleigh) and 5 m s$^{-1}$ (Mie), which are considered to
be invalid observations.





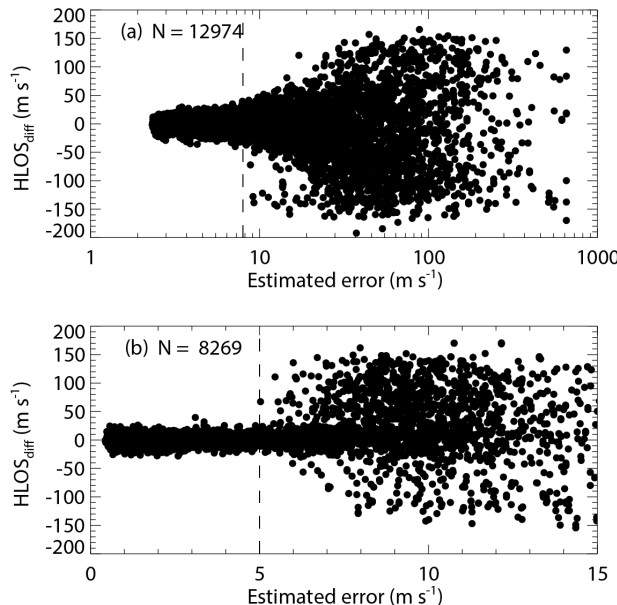

**Figure 3.** Same as Fig. 2 but for baseline 2B10.

where $N$ is the number of available data points. In addition to the STD, the scaled median absolute deviation (scaled MAD) is calculated as

$$\text{scaled MAD} = 1.4826 \times \text{median} \left( \left| HLOS_{diff}(i) - \text{median} \left( HLOS_{diff}(i) \right) \right| \right). \tag{5}$$

MAD is used as a very robust measure for the variability of the Aeolus HLOS winds because it is less sensitive to outliers than the STD (Lux et al., 2020b; Witschas et al., 2020; Baars et al., 2020; Rennie and Isaksen, 2020; Martin et al., 2021). When a data set follows a normal distribution, the MAD value multiplied by 1.4826 (scaled MAD) is identical to the STD (Ruppert and Matteson, 2015). By assuming independence between Aeolus measurements and reference instruments' measurements, the total variance of the difference between them (squared scaled MAD) ($\sigma_{val}^2$) is the sum of the variance resulting from the Aeolus random error ($\sigma_{Aeolus}^2$) and the variance resulting from reference instruments' random error ($\sigma_{i=WPR,CDWL,GPS-RS}^2$). Thus, the Aeolus random error $\sigma_{Aeolus}$ is calculated as

$$\sigma_{Aeolus} = \sqrt{\sigma_{val}^2 - \sigma_i^2}, \tag{6}$$

where $\sigma_{WPR}$, $\sigma_{CDWL}$, and $\sigma_{GPS-RS}$ are assumed to be 3, 2, and 0.7 m s$^{-1}$, respectively (see Section 2.2, 2.3, and 2.4). Note that this estimation of $\sigma_{Aeolus}$ includes the representativeness error due to the spatial and temporal mismatch between Aeolus and reference instruments' measurements. In addition to the BIAS, STD, and scaled MAD, the correlation coefficient (R) between Aeolus HLOS winds and reference instruments' HLOS winds, and the slopes and intercepts of the linear regression lines are used to evaluate the results of comparison.





## 5 Results

### 5.1 Comparison of Aeolus and WPR wind data

#### 5.1.1 Overall intercomparison

Scatterplots of Aeolus HLOS wind speed against WPR HLOS wind speed for Rayleigh-clear winds and Mie-cloudy winds during the baseline 2B02 and 2B10 periods are presented in Figs. 4 and 5, respectively. Summaries of the statistical

parameters retrieved from the scatter plot analyses for the baseline 2B02 and 2B10 are given in Tables 5 and 6, respectively. During the baseline 2B02 period, the numbers of data pairs for Rayleigh-clear and Mie-cloudy winds plotted against WPR winds are 3053 and 2687, respectively. During the baseline 2B10 period, 8443 and 6050 data pairs are provided for Rayleigh-clear and Mie-cloudy wind validation, respectively, about 2.5 times the numbers during the baseline 2B02 period. The increased number of data pairs can be explained by there being about twice as many periods for the baseline 2B10. The

energy decrease in the FM-A laser during the baseline 2B02 period led to fewer Rayleigh-clear winds that can be used for the comparison. Since 5 March 2019, Aeolus Mie-cloudy winds have been processed with a smaller horizontal averaging length of down to 10 km, also leading to more Mie-cloudy winds that can be used for comparison during the baseline 2B10

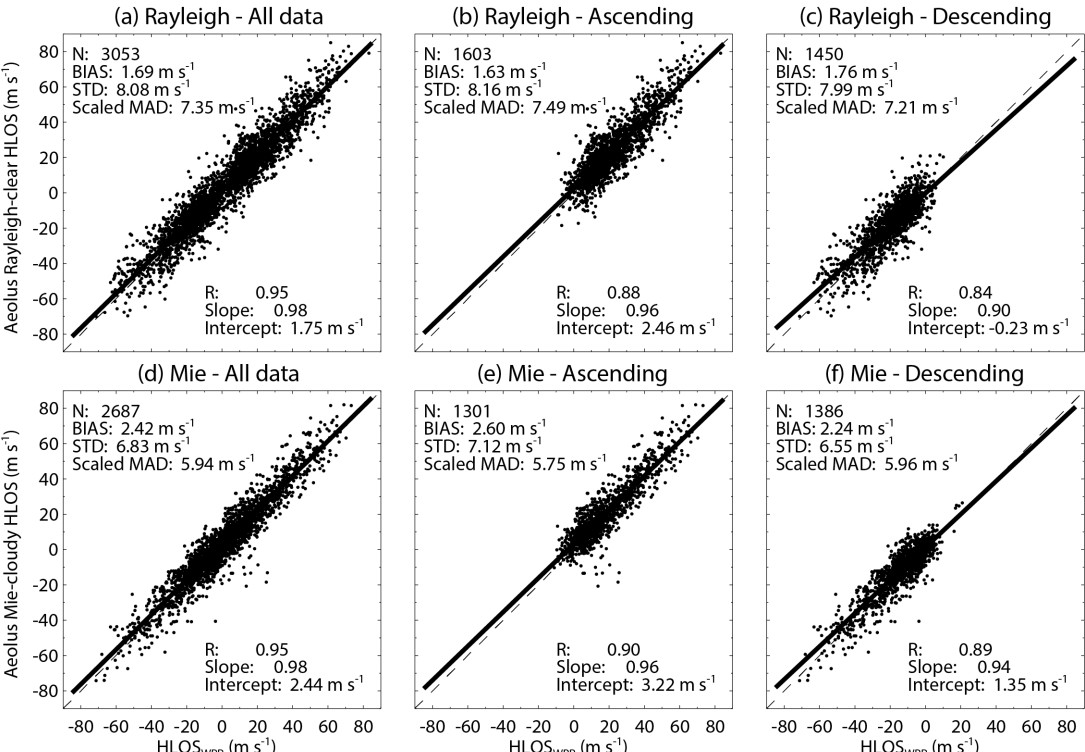

**Figure 4.** Aeolus HLOS wind speed plotted against the WPR HLOS wind speed for (a, b, c) Rayleigh-clear winds and (d, e, f) Mie-cloudy winds for (a, d) all data and (b, e) ascending and (c, f) descending orbits for baseline 2B02. Corresponding least-square line fits are indicated by the thick solid lines. The fit results are shown in the insets. The x = y line is represented by the dashed line.





period. Furthermore, the range-gate settings of Aeolus were changed on 26 February 2019, which also increased the number of available data points during the baseline 2B10 period.

During the baseline 2B02 and 2B10 periods, the linear trend between the Rayleigh-clear (Mie-cloudy) winds and WPR

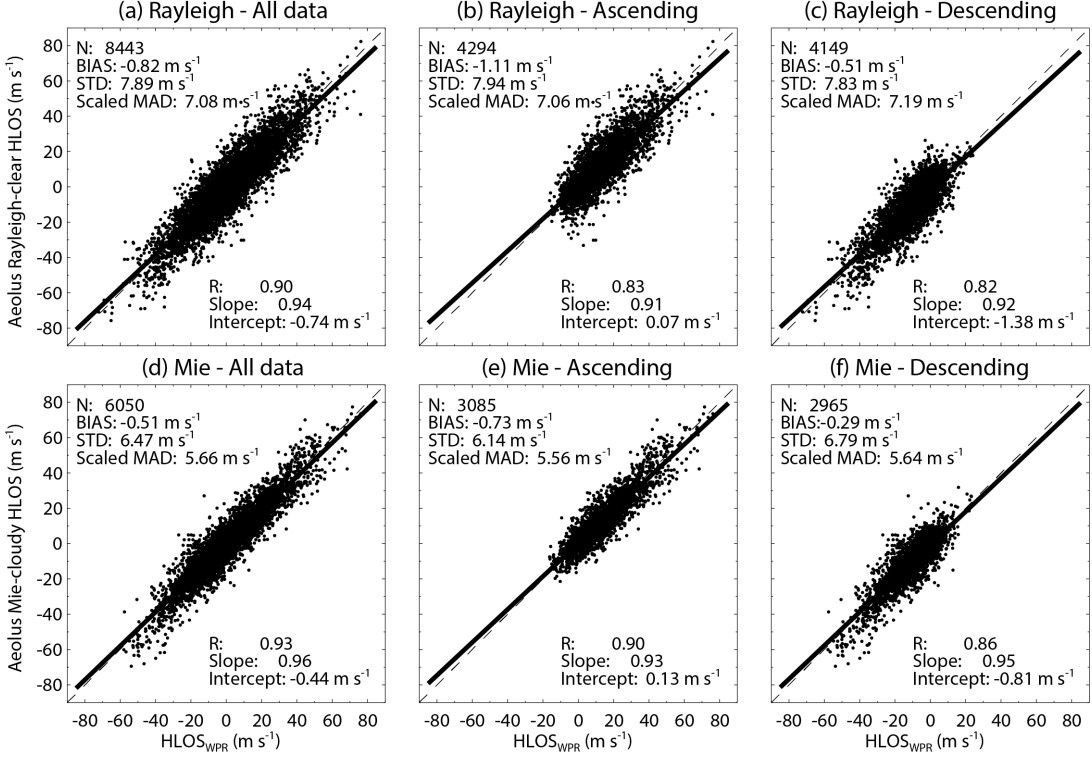

**Figure 5.** Same as Fig. 4 but for baseline 2B10.

**Table 5.** Statistical comparison of Aeolus HLOS winds and WPR HLOS winds for baseline 2B02.

| Statistical parameter | Rayleigh-clear | | | Mie-cloudy | | |
|---|---|---|---|---|---|---|
| | All data | Ascending | Descending | All data | Ascending | Descending |
| N points | 3053 | 1603 | 1450 | 2687 | 1301 | 1386 |
| BIAS (m s$^{-1}$) | 1.69 | 1.63 | 1.76 | 2.42 | 2.60 | 2.24 |
| STD (m s$^{-1}$) | 8.08 | 8.16 | 7.99 | 6.83 | 7.12 | 6.55 |
| Scaled MAD (m s$^{-1}$) | 7.35 | 7.49 | 7.21 | 5.94 | 5.75 | 5.96 |
| $\sigma_{Aeolus}$ (m s$^{-1}$) | 6.71 | 6.86 | 6.56 | 5.12 | 4.91 | 5.14 |
| Correlation | 0.95 | 0.88 | 0.84 | 0.95 | 0.90 | 0.89 |
| Slope | 0.98 | 0.96 | 0.90 | 0.98 | 0.96 | 0.94 |
| Intercept (m s$^{-1}$) | 1.75 | 2.46 | −0.23 | 2.44 | 3.22 | 1.35 |



winds is clearly seen for all data and both orbit phases (Figs. 4 and 5). Although the Rayleigh-clear winds for all data and both orbit phases exhibit a slightly positive bias between 1.6 and 1.8 m s$^{-1}$ during the baseline 2B02 period (Figs. 4a–c), no significant wind-speed-dependent bias is apparent. However, the systematic errors (biases) obtained in this study are higher than those of 0.7 m s$^{-1}$ stipulated in the mission requirements (Ingmann and Straume, 2016). The slopes of the linear
regression line of Rayleigh-clear versus WPR winds are 0.98, 0.96, and 0.90 for all data, the ascending orbit, and the descending orbit, respectively. High correlation coefficients were also found: 0.95 for all data, 0.88 for the ascending orbit, and 0.84 for the descending orbit. That is, the slopes of the fit are not significantly different from 1 and the correlation coefficients exceed 0.8. The random error represented by the scaled MAD was determined to be 7.21–7.49 m s$^{-1}$ for the Rayleigh-clear winds. Lux et al. (2020b) compared the Rayleigh-clear winds with winds measured with the ALADIN
Airborne Demonstrator (A2D) during the WindVal III validation campaign carried out in central Europe from 17 November to 5 December 2018 (i.e., during the baseline 2B02 period). They reported a bias of 2.56 m s$^{-1}$ with a scaled MAD of 3.57 m s$^{-1}$. They also reported that the slope of the linear regression line and the correlation coefficient of Rayleigh-clear versus A2D winds were 0.83 and 0.80, respectively. Witschas et al. (2020) reported a bias of 2.11 m s$^{-1}$ with a scaled MAD of 3.97 m s$^{-1}$ for Rayleigh-clear winds during the same campaign (WindVal III) using an airborne 2 μm CDWL. They also reported
that the slope of the linear regression line and the correlation coefficient were 0.99 and 0.95, respectively. Thus, the bias, slope, and correlation coefficient of Rayleigh-clear versus WPR winds are consistent with those derived from other Aeolus validation campaigns, but the scaled MAD is significantly larger. The large scaled MAD leads to a large $\sigma_{Aeolus}$ (6.56–6.86 m s$^{-1}$). The Aeolus random error of Rayleigh-clear winds is significantly larger than the 2.5 m s$^{-1}$ stipulated in the mission requirements at 2−16 km altitude (Ingmann and Straume, 2016). Witschas et al. (2020) estimated a $\sigma_{Aeolus}$ of 3.9 m s$^{-1}$ for
Rayleigh-clear winds by excluding the 2 μm CDWL measurement error during the commissioning phase. This discrepancy is probably related to the large representativeness error due to the large sampling volume of the WPR.

During the baseline 2B10 period, the biases of Rayleigh-clear winds are slightly negative (–0.82, –1.11, and –0.51 m s$^{-1}$) for all data and both orbit phases (Figs. 5a–c). The absolute values of the biases during the baseline 2B10 period are about half of those during the baseline 2B02 period. The slightly negative biases are generally consistent with those reported by Guo et
al. (2021), who compared the Rayleigh-clear winds with winds measured with the radar wind profiler network in China from 20 April to 20 July 2020. The slopes of the linear regression line (correlation coefficients) of Rayleigh-clear versus WPR winds are 0.94 (0.90), 0.91 (0.83), and 0.92 (0.82) for all data, the ascending orbit, and the descending orbit, respectively. These values are almost the same as those of the baseline 2B02 and agree well with those reported by Guo et al. (2021). The scaled MADs (7.06–7.19 m s$^{-1}$) are marginally smaller than those of the baseline 2B02. Although the random error is
significantly large, these results indicate that the Aeolus Rayleigh-clear winds are broadly consistent with WPR winds over Japan.





**Table 6.** Same as Table 5 but for baseline 2B10.

| Statistical parameter | Rayleigh-clear | | | Mie-cloudy | | |
|---|---|---|---|---|---|---|
| | All data | Ascending | Descending | All data | Ascending | Descending |
| N points | 8443 | 4294 | 4149 | 6050 | 3085 | 2965 |
| BIAS (m s$^{-1}$) | −0.82 | −1.11 | −0.51 | −0.51 | −0.73 | −0.29 |
| STD (m s$^{-1}$) | 7.89 | 7.94 | 7.83 | 6.47 | 6.14 | 6.79 |
| Scaled MAD (m s$^{-1}$) | 7.08 | 7.06 | 7.19 | 5.66 | 5.56 | 5.64 |
| $\sigma_{Aeolus}$ (m s$^{-1}$) | 6.42 | 6.39 | 6.54 | 4.80 | 4.68 | 4.77 |
| Correlation | 0.90 | 0.83 | 0.82 | 0.93 | 0.90 | 0.86 |
| Slope | 0.94 | 0.91 | 0.92 | 0.96 | 0.93 | 0.95 |
| Intercept (m s$^{-1}$) | −0.74 | 0.07 | −1.38 | −0.44 | 0.13 | −0.81 |

The same statistics are shown for the Mie-cloudy winds in Figs. 4d–f and 5d–f. The biases of Mie-cloudy versus WPR winds are positive for all data and both orbit phases (2.42, 2.60, and 2.24 m s$^{-1}$) during the baseline 2B02 period (Figs. 4d–f). The biases are beyond the mission requirements of Aeolus and slightly larger than the Rayleigh-clear bias (Figs. 4a–c). The
slopes of the linear regression line (correlation coefficients) are 0.98 (0.95), 0.96 (0.90), and 0.94 (0.89) for all data, the ascending orbit, and the descending orbit, respectively. As with the Rayleigh-clear winds, the slopes of the fit are not significantly different from 1 and correlation coefficients exceed 0.8. The scaled MAD is determined to be 5.75–5.96 m s$^{-1}$ and slightly smaller than that of the Rayleigh-clear winds. Witschas et al. (2020) reported a bias of 2.26 m s$^{-1}$ with a scaled MAD of 2.22 m s$^{-1}$ for Mie-cloudy winds during the WindVal III validation campaign. The slope of the linear regression
line (correlation coefficient) was 0.96 (0.92). Therefore, the bias, slope, and correlation coefficient of Mie-cloudy versus WPR winds derived in this study are almost the same as the results of Witschas et al. (2020), but the random error is significantly larger. The large scaled MAD leads to a large $\sigma_{Aeolus}$ (4.91–5.14 m s$^{-1}$). Witschas et al. (2020) estimated a $\sigma_{Aeolus}$ of 2.0 m s$^{-1}$ for Mie-cloudy winds by excluding the 2 μm CDWL measurement error during the commissioning phase. This discrepancy may be caused by the larger representativeness error due to the large sampling volume of the WPR.
The same statistics are shown for the baseline 2B10 in Figs. 5d–f. For all data, the slope of the linear regression line and the correlation coefficient for the Mie-cloudy winds are 0.96 and 0.93, respectively. These values are almost the same as those of the Rayleigh-clear winds. The slopes of the linear regression line (correlation coefficient) are 0.93 (0.90) and 0.95 (0.86) for ascending and descending orbits, respectively. These results indicate that the performance of Aeolus for Mie-cloudy winds is reliable over Japan. The biases of Mie-cloudy versus WPR winds are slightly negative for all data and both orbit
phases (−0.51, −0.73, and −0.29 m s$^{-1}$), but these values are smaller than those of the Rayleigh-clear winds. As with the Rayleigh-clear winds, the absolute bias is slightly larger for the ascending orbit than for the descending orbit. The small bias,





slope close to 1, and high correlation coefficient agree well with those reported by Guo et al. (2021). The scaled MADs are relatively large (5.56–5.66 m s$^{-1}$), but the values are smaller than those of the Rayleigh-clear winds.

To summarize, the systematic and random errors of Rayleigh-clear (Mie-cloudy) versus WPR winds for the baseline 2B10

are improved as compared with those for the baseline 2B02. In contrast to the baseline 2B02, the systematic error of Mie-cloudy winds is superior to that of Rayleigh-clear winds during the baseline 2B10 period. During the baseline 2B02 period, the systematic error was significantly larger than the strict mission requirement of 0.7 m s$^{-1}$ specified for Aeolus HLOS winds. During the baseline 2B10 period, both Rayleigh-clear and Mie-cloudy winds have generally met the mission requirements on systematic errors. However, the Aeolus random error of Rayleigh-clear and Mie-cloudy winds is

considerably larger than the required precision of 2.5 m s$^{-1}$ in the free troposphere during the baseline 2B02 and 2B10 periods. The main reason for not yet achieving the mission requirement for random errors is probably related to the large representativeness error due to the large sampling volume of the WPR. From the statistical comparisons, we found no significant difference between the ascending and descending orbits with respect to the Rayleigh-clear and Mie-cloudy winds during the baseline 2B02 and 2B10 periods.


### 5.1.2 Vertical distribution of wind differences

The vertical distributions of the bias and standard deviation of the differences between Aeolus and WPR HLOS winds for baseline 2B02 are shown in Fig. 6. The values are binned into bins of 1 km height. The bias uncertainties estimated at 90 % confidence level for all data are reasonably small up to about 10 km altitude where there are relatively many paired data

points for comparison (Fig. 6a). For all data, the biases of Rayleigh-clear and WPR HLOS winds are significantly positive in all altitude ranges and less than 3.53 m s$^{-1}$ up to 10 km. The larger standard deviations at 0−2 km altitude for ascending and descending orbits are caused by fewer paired data points. For Mie-cloudy winds, the biases for all data are also significantly positive in all altitude ranges except for 10–11 km (Fig. 6d). Although the biases are also positive below 8 km during ascending and descending orbits, the vertical distributions of bias are opposite to each other above 8 km. The mission

requirement of 0.7 m s$^{-1}$ has not been achieved by both Rayleigh-clear and Mie-cloudy biases in all altitude ranges.



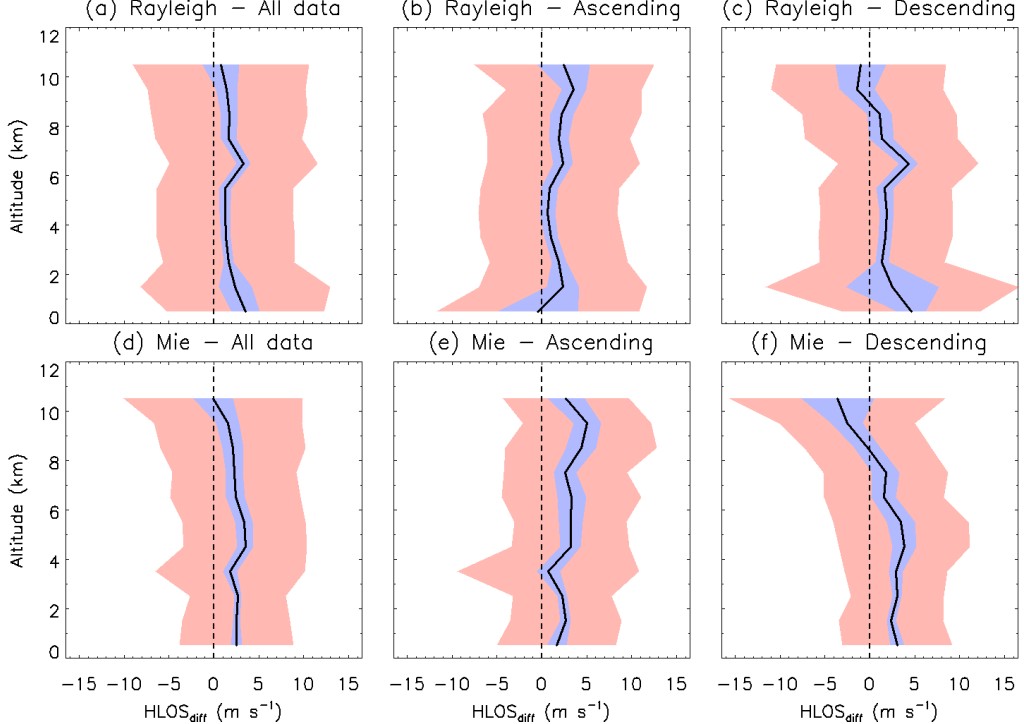

**Figure 6.** Vertical profiles in 1 km bins of the HLOS wind speed differences between the Aeolus and WPR HLOS winds for (a, b, c) Rayleigh-clear winds and (d, e, f) Mie-cloudy winds for (a, d) all data and (b, e) ascending and (c, f) descending orbits for baseline 2B02. Thick black lines show the bias with the blue shaded areas corresponding to the 90% confidence interval. The red shaded areas represent 1 standard deviation on each side of the bias.

The same statistics are shown for the baseline 2B10 in Fig. 7. As with the baseline 2B02, the bias uncertainties estimated at 90 % confidence level are reasonably small up to about 11 km altitude where there are relatively many paired data points for comparison. For all data, the biases of Rayleigh-clear and WPR HLOS winds are slightly negative in all altitude ranges and less than $-1.6$ m s$^{-1}$ up to 11 km (Fig. 7a). The systematic error was less than that of the baseline 2B02. Below 2 km altitude,

the Rayleigh-clear winds have met the mission requirements for systematic errors. The bias and standard deviation in the altitude range of 0–1 km (atmospheric boundary layer) are almost the same as those in the upper level. This suggests that the correction scheme against the scattering ratio was improved in the L2B processor (see Section 5.1.4). However, this result is different from that in the other validation studies conducted during the baseline 2B10 period (Guo et al., 2021). For the ascending (descending) orbit, the minimum (maximum) bias is $-2.34$ (0.56) m s$^{-1}$ in the altitude range of 5–6 (6–7) km. The

vertical distributions of bias during ascending and descending orbits are opposite to each other in the altitude range of 3–11 km. For all data, the biases of Mie-cloudy and WPR HLOS winds are also slightly negative in all altitude ranges except for 3–4 km (Fig. 7d). As with the Rayleigh-clear winds, the systematic error was improved as compared with that of the baseline 2B02. Below 5 km altitude, Mie-cloudy winds have met the mission requirements on systematic errors. For the ascending



(descending) orbit, the minimum (maximum) bias is –1.93 (0.54) m s$^{-1}$ in the altitude range of 5–6 (4–5) km. As with the

Rayleigh-clear winds, the vertical distributions of bias during ascending and descending orbits are opposite to each other in the altitude range of 3–11 km.

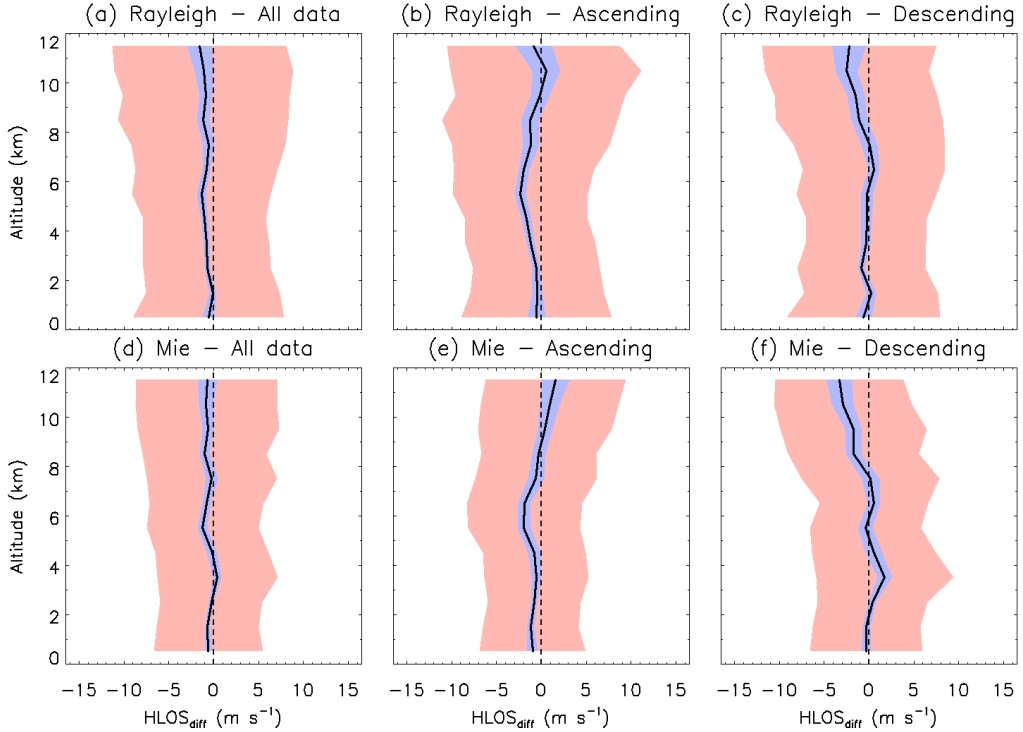

**Figure 7.** Same as Fig. 6 but for baseline 2B10.

### 5.1.3 Time series variation of wind differences

The time series variation of the bias and standard deviation of the differences between Aeolus and WPR HLOS winds during

the baseline 2B02 period are shown in Fig. 8. Immediately after the launch of Aeolus, the biases of the Rayleigh-clear and Mie-cloudy winds are small for all data and both orbit phases. With time, the Rayleigh-clear and Mie-cloudy biases increase for all data and both orbit phases. The Rayleigh-clear bias reached its maximum in January 2019. For the Mie-cloudy winds, the maximums occurred in January and February 2019 for ascending and descending orbits, respectively. Within the baseline 2B02, the Rayleigh-clear and Mie-cloudy biases show a positive trend.

For the baseline 2B10, the same statistics are shown in Fig. 9. For all data, the biases of Rayleigh-clear and WPR HLOS winds are generally negative at all months except August 2020, but the biases do not show a significant seasonal trend (Fig. 9a). The standard deviations of Rayleigh-clear and WPR HLOS data gradually increase with time (from 6.34 to 8.77 m s$^{-1}$).

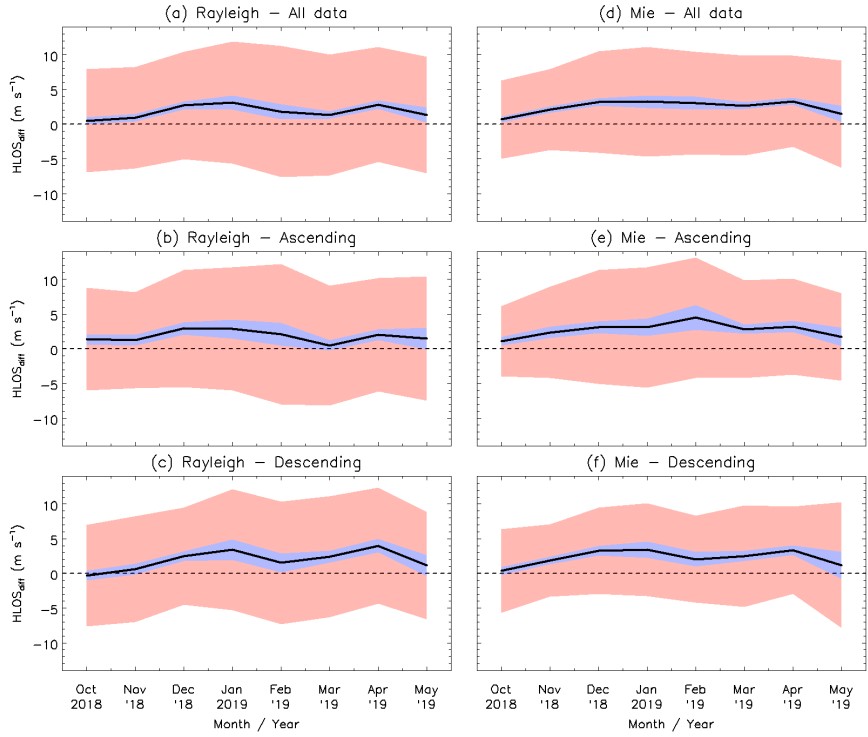

**Figure 8.** Monthly averaged values of wind speed differences between the Aeolus and WPR HLOS winds for (a, b, c) Rayleigh-clear winds and (d, e, f) Mie-cloudy winds for (a, d) all data and (b, e) ascending and (c, f) descending orbits for baseline 2B02. Thick black lines show the bias with the blue shaded areas corresponding to the 90% confidence interval. The red shaded areas represent 1 standard deviation on each side of the bias.

A possible reason is the decrease in the level of the received signal after passing through the telescope. The biases for the ascending orbit are negative throughout the whole period (Fig. 9b). The absolute biases are generally larger for the ascending orbit than for the descending orbit (Figs. 9b and 9c). For all data, the biases of Mie-cloudy and WPR HLOS winds gradually fluctuate and do not show a significant seasonal trend (Fig. 9c). The bias and standard deviation of Mie-cloudy winds are generally smaller than those of Rayleigh-clear winds. There is no significant increase in the standard deviations of Mie-cloudy winds with time, because the Mie return signal does not depend on the laser energy (Martin et al., 2021). It is interesting to note that the fluctuation of the bias was stronger for the descending orbit than for the ascending orbit in 2019 (Figs. 9e and 9f). However, the biases for both orbit phases approached zero in 2020.





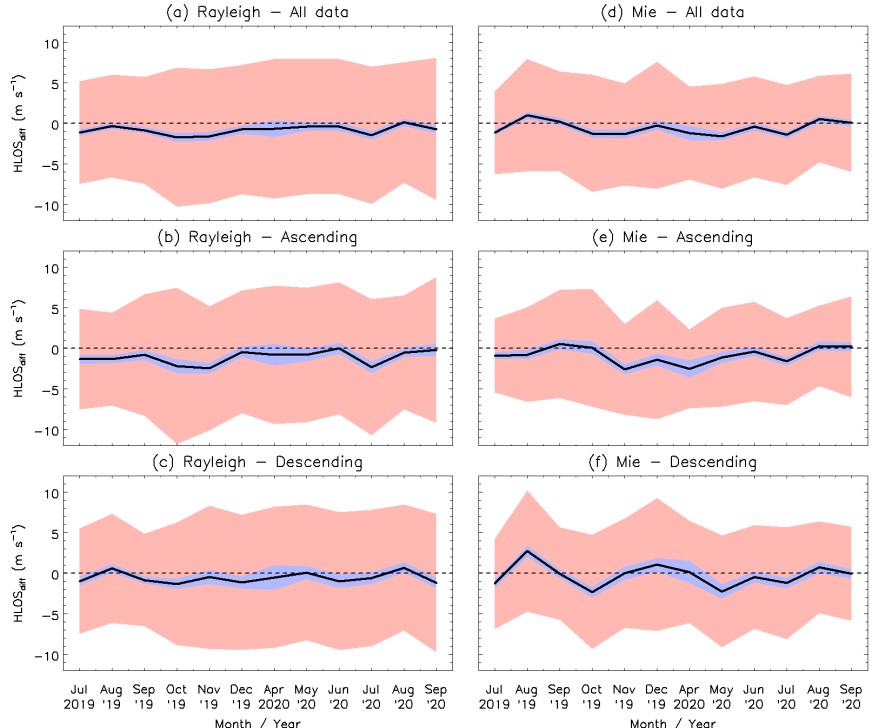

**Figure 9.** Same as Fig. 8 but for baseline 2B10.

### 5.1.4 Rayleigh-clear wind bias dependence on scattering ratio

The scattering ratio on the Rayleigh channel is defined as the ratio of the total scattering signal (particles and molecules) to the molecular scattering signal. When the scattering ratio is large, a strong narrowband Mie return signal partly enters the

Rayleigh spectrometer, changing the sensitivity of the Rayleigh channel (Witschas et al. 2020). The dependence of the Rayleigh-clear wind bias on the scattering ratio given in the L2B product is shown in Fig. 10. It can be seen that the scattering ratio varied between 1.1 and 1.4 for baseline 2B02 and between 1.05 and 1.65 for baseline 2B10. This means that the determination of the scattering ratio and the threshold for classifying the Rayleigh-clear winds changed between the baselines 2B02 and 2B10. During the baseline 2B02 period, the bias of Rayleigh-clear and WPR HLOS winds slightly

increased as the scattering ratio increased (Fig. 10a). Using the L2B products within the commissioning phase, Witschas et al. (2020) reported that the scattering ratio has a considerable influence on the bias of Rayleigh-clear winds. During the baseline 2B10 period, the Rayleigh-clear winds exhibited a slightly negative bias and there was no significant bias dependence on the scattering ratio (Fig. 10b). This means that the correction scheme of the scattering ratio was improved in the L2B processor.

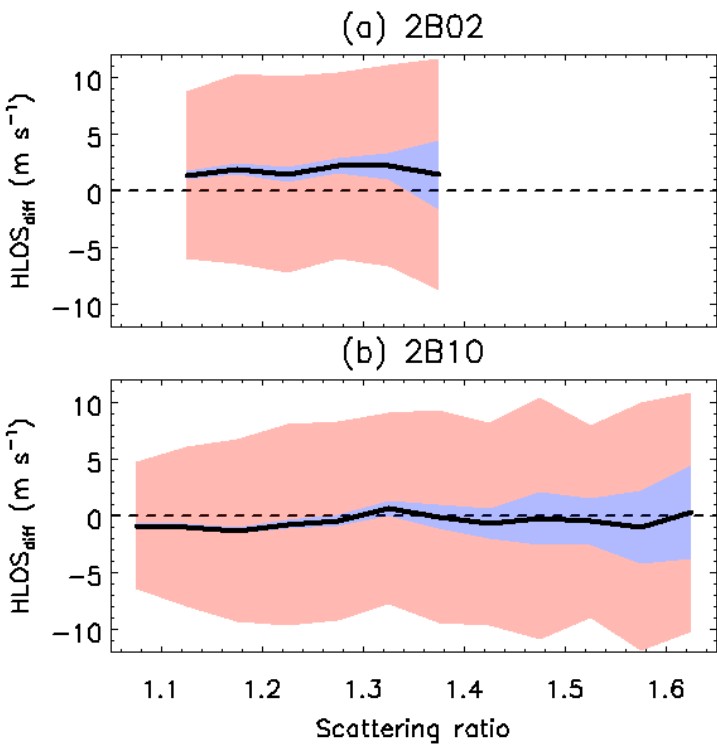

**Figure 10.** Dependence of wind speed differences between the Aeolus Rayleigh-clear and WPR HLOS winds on scattering ratio during the baseline (a) 2B02 and (b) 2B10 periods.

## 5.2 Comparison of Aeolus and CDWL wind data

Scatterplots of Aeolus HLOS winds against CDWL HLOS winds for Rayleigh-clear and Mie-cloudy winds during the baseline 2B02 period are presented in Fig. 11. Summaries of the statistical parameters retrieved from the scatter plot analysis for the baseline 2B02 and 2B10 are given in Table 7. While Okinawa is located at the southern edge of the subtropical jet stream, Kobe is located just below the subtropical jet stream. Thus, the CDWL at Kobe sampled a higher wind speed of the subtropical jet stream. It can be seen that the acquired HLOS wind speed range was wider for Kobe than for Okinawa in Fig. 11. Both Rayleigh-clear and Mie-cloudy winds exhibit a slightly positive bias. The different colors indicate whether Aeolus had an ascending orbit (red) or descending orbit (blue). There is no significant difference between the ascending and descending orbits. The slopes of the linear regression lines are 1.05 (Rayleigh) and 1.05 (Mie) at Kobe and 0.99 (Rayleigh) and 1.01 (Mie) at Okinawa. The correlation coefficients are 0.98 (Rayleigh) and 0.98 (Mie) at Kobe and 0.93 (Rayleigh) and 0.97 (Mie) at Okinawa. That is, the slopes of the fit and the correlation coefficients of Rayleigh-clear and Mie-cloudy winds are not significantly different from 1 at Kobe and Okinawa. The intercepts of the linear regression lines are determined to be 0.61 m s$^{-1}$ (Rayleigh) and 1.76 m s$^{-1}$ (Mie) at Kobe and 1.07 m s$^{-1}$ (Rayleigh) and 2.37 m s$^{-1}$ (Mie) at Okinawa. A similar

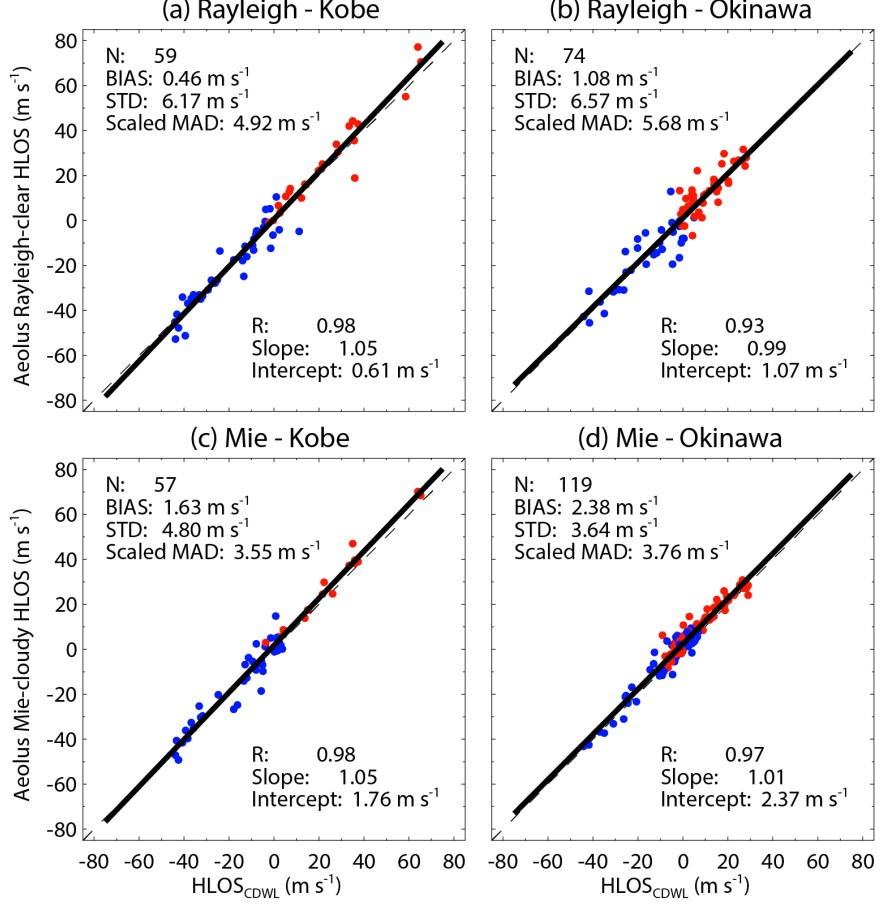

**Figure 11.** Aeolus against CDWL HLOS winds for (a, b) Rayleigh-clear winds and (c, d) Mie-cloudy winds at (a, c) Kobe and (b, d) Okinawa for baseline 2B02. Corresponding least-square line fits are indicated by the thick solid lines. The fit results are shown in the insets. The x = y line is represented by the dashed line. Red circles represent measurements of an ascending orbit, whereas blue circles represent measurements of a descending orbit.

finding is obtained from the biases that are 0.46 m s$^{-1}$ (Rayleigh) and 1.63 m s$^{-1}$ (Mie) at Kobe and 1.08 m s$^{-1}$ (Rayleigh) and 2.38 m s$^{-1}$ (Mie) at Okinawa. Both Rayleigh-clear and Mie-cloudy winds exhibit a slightly positive bias. Except for

Rayleigh-clear winds measured at Kobe, the systematic error has not achieved the mission requirement of 0.7 m s$^{-1}$. This result is similar to that in the comparisons of Aeolus and WPR measurements. The systematic error of CDWL observations is smaller than 0.2 m s$^{-1}$ (see Section 2.3) and thus does not significantly contribute to the biases here. The random errors represented by the scaled MADs of Rayleigh-clear (Mie-cloudy) winds are 4.92 (3.55) m s$^{-1}$ at Kobe and 5.68 (3.76) m s$^{-1}$ at Okinawa. The values are smaller than the scaled MADs of Rayleigh-clear (Mie-cloudy) versus WPR winds. The $\sigma_{Aeolus}$ of

Rayleigh-clear (Mie-cloudy) winds is determined using Eq. 5 to be 4.49 (2.93) m s$^{-1}$ at Kobe and 5.31 (3.19) m s$^{-1}$ at Okinawa. Witschas et al. (2020) determined $\sigma_{Aeolus}$ of 3.9 m s$^{-1}$ for Rayleigh-clear winds and 2.0 m s$^{-1}$ for Mie-cloudy



winds by excluding the airborne 2 μm CDWL measurement error during the commissioning phase. The discrepancies are probably caused by the smaller representativeness error due to the spatial and temporal displacements between Aeolus and airborne CDWL measurements.

**Table 7.** Statistical comparison of Aeolus HLOS winds and CDWL HLOS winds.

| Baseline | 2B02 | | | | 2B10 | | | |
|---|---|---|---|---|---|---|---|---|
| Site | Kobe | | Okinawa | | Kobe | | Okinawa | |
| Rayleigh/Mie | Rayleigh | Mie | Rayleigh | Mie | Rayleigh | Mie | Rayleigh | Mie |
| N points | 59 | 57 | 74 | 119 | 204 | 136 | 232 | 220 |
| BIAS (m s$^{-1}$) | 0.46 | 1.63 | 1.08 | 2.38 | –0.81 | 0.16 | –0.48 | –0.26 |
| STD (m s$^{-1}$) | 6.17 | 4.80 | 6.57 | 3.64 | 5.69 | 5.15 | 6.53 | 4.74 |
| Scaled MAD (m s$^{-1}$) | 4.92 | 3.55 | 5.68 | 3.76 | 5.21 | 3.92 | 5.58 | 3.86 |
| $\sigma_{Aeolus}$ (m s$^{-1}$) | 4.49 | 2.93 | 5.31 | 3.19 | 4.81 | 3.37 | 5.21 | 3.30 |
| Correlation | 0.98 | 0.98 | 0.93 | 0.97 | 0.96 | 0.97 | 0.79 | 0.86 |
| Slope | 1.05 | 1.05 | 0.99 | 1.01 | 0.98 | 1.02 | 1.03 | 0.86 |
| Intercept (m s$^{-1}$) | 0.61 | 1.76 | 1.07 | 2.37 | –0.88 | 0.22 | –0.52 | –0.04 |

Figure 12 shows the correlation plots of the Aeolus HLOS winds against CDWL HLOS winds for Rayleigh-clear and Mie-cloudy winds at Kobe and Okinawa during the baseline 2B10 period. As with the baseline 2B02 period, a linear trend between Aeolus and CDWL measurements is clearly seen from the linear regression. At Kobe, the correlation coefficients are 0.96 and 0.97 for Rayleigh-clear and Mie-cloudy winds, respectively, and close to 1. At Okinawa, the correlation coefficients are 0.79 and 0.86 for Rayleigh-clear and Mie-cloudy winds, respectively, and are smaller than those at Kobe. At

Okinawa, 47% and 62% of the data pairs for Rayleigh-clear and Mie-cloudy winds versus CDWL winds are obtained below 2 km altitude, respectively. This result is suggested to be linked to the strong convection in the atmospheric boundary layer at Okinawa, especially in summer. The intercepts of the linear regression lines are determined to be –0.88 m s$^{-1}$ (Rayleigh) and 0.22 m s$^{-1}$ (Mie) at Kobe and –0.52 m s$^{-1}$ (Rayleigh) and –0.04 m s$^{-1}$ (Mie) at Okinawa. The biases are –0.81 m s$^{-1}$ (Rayleigh) and 0.16 m s$^{-1}$ (Mie) at Kobe and –0.48 m s$^{-1}$ (Rayleigh) and –0.26 m s$^{-1}$ (Mie) at Okinawa. The absolute bias of

Rayleigh-clear and Kobe CDWL winds is slightly larger than that for the baseline 2B02, the reason for which is unclear. Except for Rayleigh-clear winds measured at Kobe, the systematic error has achieved the mission requirement of 0.7 m s$^{-1}$. The scaled MADs of Rayleigh-clear (Mie-cloudy) winds are 5.21 (3.92) m s$^{-1}$ at Kobe and 5.58 (3.86) m s$^{-1}$ at Okinawa. In contrast to the comparisons of Aeolus and WPR measurements, the random errors are almost the same as those for the baseline 2B02, and no improvement of the random error is evident. As with the scaled MADs, the estimated $\sigma_{Aeolus}$ of

Rayleigh-clear and Mie-cloudy winds at Kobe and Okinawa are almost the same as those for the baseline 2B02.

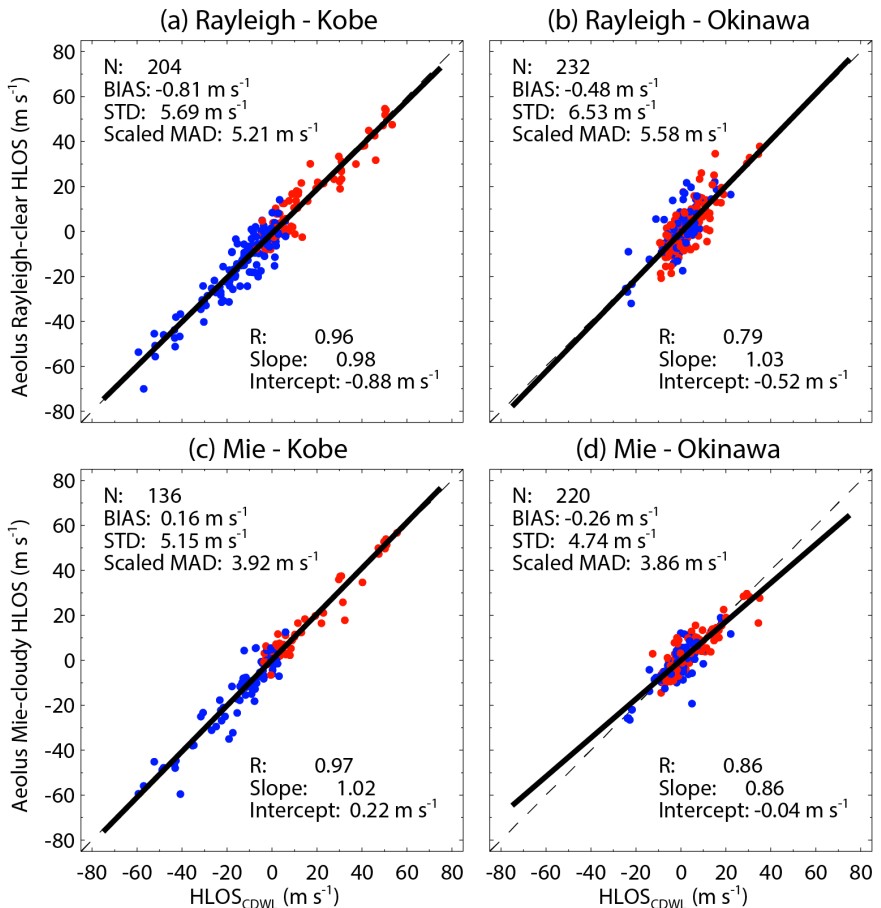

**Figure 12.** Same as Fig. 11 but for baseline 2B10.

## 5.3 Comparison of Aeolus and GPS-RS wind data

For the validation of the Aeolus wind products, we launched 12 and 6 GPS-RSs from NICT Okinawa during the baseline
2B02 and 2B10 periods, respectively (Table 4). The GPS-RSs obtained wind profiles with a vertical range up to 25 km. Thus,

the GPS-RSs can measure winds of the upper troposphere and lower stratosphere which cannot be measured by the WPRs
and CDWLs.

Figure 13a shows HLOS wind speed profiles measured by the GPS-RSs with the Rayleigh-clear and Mie-cloudy profiles on
8 November 2018. The Rayleigh-clear winds show good coverage and closely follow the shape of the wind profile at
altitudes higher than 2 km. The subtropical jet stream with westerly winds can be seen in the GPS-RS and Rayleigh-clear

observations at around 14 km. A maximum absolute wind speed higher than 50 m s$^{-1}$ was observed in this height region





according to the high-resolution GPS-RS profile. Despite the coarse range resolution (2 km) of the Aeolus measurements in this height region, the Rayleigh-clear winds are able to detect the high wind speed. The Mie-cloudy winds are available below 4.5 km and at high altitudes of 9–11.5 km owing to the occurrence of cirrus clouds. A cirrus cloud layer was also observed by the CDWL during the overpass of Aeolus (not shown). There are large deviations between Mie-cloudy and

GPS-RS winds below 2 km. Since the horizontal distance between the Mie-cloudy measurements and the GPS-RS is about 100 km in this height region, one can assume that the reason for the large deviations is the spatial heterogeneity of the horizontal wind in the atmospheric boundary layer.

The second case discussed in this study is from 1 December 2018 (Fig. 13b). The Mie-cloudy winds are available below 4 km. Since the occurrence of cloud was sporadically detected by the CDWL at 3−4 km altitude (not shown), it is assumed that

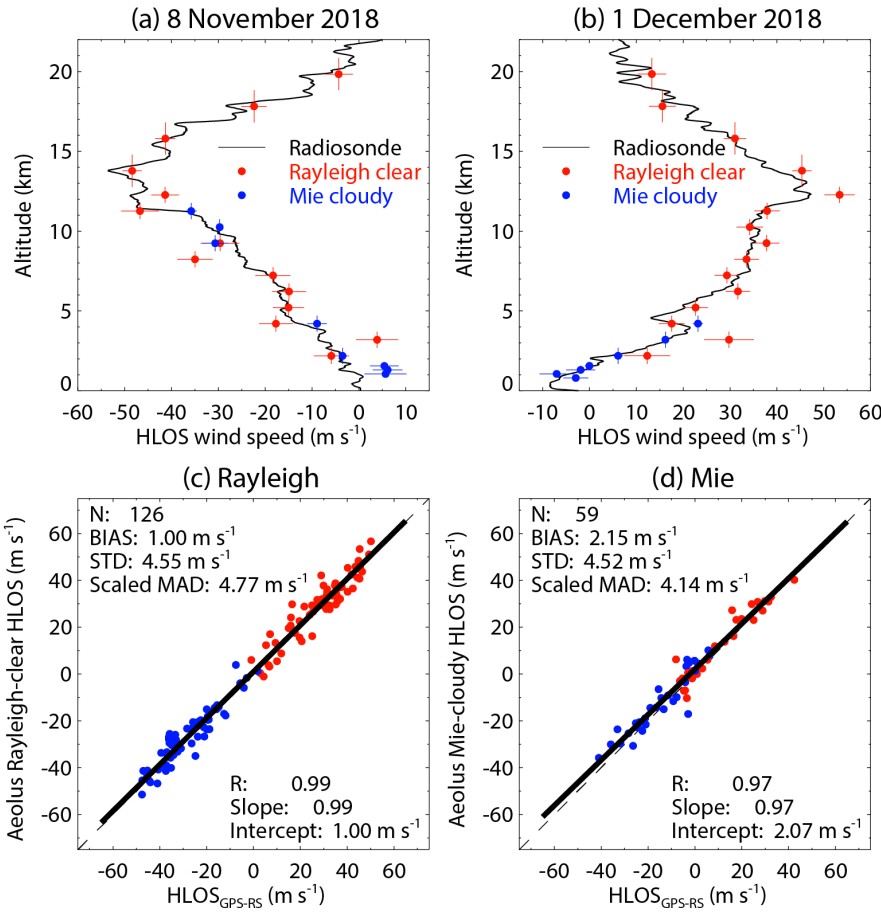

**Figure 13.** (a) HLOS wind speed profiles measured by the GPS-RS (thin black line) with the Rayleigh-clear (red) and Mie-cloudy (blue) profiles on 8 November 2018. (b) Same as Fig. 13a but on 1 December 2018. (c) Rayleigh-clear and (d) Mie-cloudy HLOS winds versus the radiosonde measurements for baseline 2B02. Red circles represent measurements of an ascending orbit, whereas blue circles represent measurements of a descending orbit.



the clouds were partly existent in the Aeolus observational domain. As compared with the previous case (8 November 2018), the Mie-cloudy winds agree with the GPS-RS winds in the lowermost 2 km. The reason for the agreement is that the Aeolus ground track was relatively near the radiosonde launching position (about 50 km). The Rayleigh-clear winds are available at altitudes higher than 2 km. As with the previous case, the subtropical jet stream with westerly winds is seen in the GPS-RS and Rayleigh-clear observations at around 12 km altitude. The Rayleigh-clear wind measurements can detect the high wind

speed, but they are slightly overestimated, the reason for which is unclear. There is a possibility regarding horizontal wind gradients in this height region.

Figures 13c and 13d show the correlation plots of the Rayleigh-clear and Mie-cloudy HLOS winds against GPS-RS HLOS winds during the baseline 2B02 period, respectively. Summaries of the statistical parameters retrieved from the scatter plot analysis for the baseline 2B02 and 2B10 are given in Table 8. A linear trend between Aeolus and GPS-RS measurements is

clearly seen from the linear regression. The linear regression line has a slope of 0.99 (0.97) with an intercept of 1.00 (2.07) m s$^{-1}$ for the comparison of Rayleigh-clear (Mie-cloudy) and GPS-RS winds. Both Rayleigh-clear and Mie-cloudy winds exhibit a slightly positive bias. The different colors indicate whether Aeolus had an ascending orbit (red) or a descending orbit (blue). No significant difference was found between the ascending and descending orbits. The biases of Rayleigh-clear and Mie-cloudy winds are 1.00 and 2.15 m s$^{-1}$, respectively. These values are almost the same as the intercept of the linear

regression line. The random error represented by the scaled MAD of Rayleigh-clear winds (4.77 m s$^{-1}$) is slightly larger than that of Mie-cloudy winds (4.14 m s$^{-1}$). Baars et al. (2020) compared the Rayleigh-clear and Mie-cloudy winds with winds obtained from the radiosonde launches onboard the German RV *Polarstern* during cruise PS116 carried out in the Atlantic Ocean west of the African continent from 17 November to 10 December 2018 (i.e., during the baseline 2B02 period). They

**Table 8.** Statistical comparison of Aeolus HLOS winds and GPS-RS HLOS winds.

| Baseline | 2B02 | | 2B10 | |
|---|---|---|---|---|
| Rayleigh/Mie | Rayleigh | Mie | Rayleigh | Mie |
| N points | 126 | 59 | 92 | 43 |
| BIAS (m s$^{-1}$) | 1.00 | 2.15 | 0.45 | −0.71 |
| STD (m s$^{-1}$) | 4.55 | 4.52 | 4.43 | 5.81 |
| Scaled MAD (m s$^{-1}$) | 4.77 | 4.14 | 3.97 | 3.99 |
| $\sigma_{Aeolus}$ (m s$^{-1}$) | 4.71 | 4.08 | 3.91 | 3.92 |
| Correlation | 0.99 | 0.97 | 0.99 | 0.95 |
| Slope | 0.99 | 0.97 | 1.01 | 0.92 |
| Intercept (m s$^{-1}$) | 1.00 | 2.07 | 0.38 | −0.22 |



reported biases of 1.52 and 0.95 m s⁻¹ with random errors of 4.84 and 1.58 m s⁻¹ for Rayleigh-clear and Mie-cloudy winds,
respectively. The slope and intercept of the linear regression line were 0.97 (0.95) and 1.57 (1.13) m s⁻¹ for the comparison
of Rayleigh-clear (Mie-cloudy) and radiosonde winds, respectively. Therefore, the slightly positive biases of Rayleigh-clear
and Mie-cloudy versus GPS-RS winds obtained in this study are almost the same as those obtained by Baars et al. (2020).
The result that the random error of Mie-cloudy winds is much smaller than that of Rayleigh-clear wind contrasts with our
results.

Figures 14a and 14b show HLOS wind speed profiles measured by the GPS-RS with the Rayleigh-clear and Mie-cloudy
profiles on 19 December 2019 and 21 December 2019, respectively. The range bins were changed to a resolution of 1 km up
to an altitude of 19 km on 26 February 2019. Owing to the high range resolution, the Rayleigh-clear wind measurements of
Aeolus can detect the rapid changes in the wind speed profiles in the subtropical jet stream. On 19 December 2019, the
CDWL observed a cloud layer at around 1 km under rainy conditions during the overpass of Aeolus (not shown), and the
Mie-cloudy winds were detected above the cloud layer. On 21 December 2019, the CDWL observed multiple cloud layers
up to about 9 km during the overpass of Aeolus (not shown), and the Mie-cloudy winds were detected at these cloud layers.
Figures 14c and 14d show the correlation plots of the Rayleigh-clear and Mie-cloudy HLOS winds against GPS-RS HLOS
winds during the baseline 2B10 period, respectively. As with the baseline 2B02, a linear trend between Aeolus and GPS-RS
observations is clearly seen from the linear regression. The linear regression line has a slope of 1.01 (0.92) with an intercept
of 0.38 (–0.22) m s⁻¹ for the comparison of Rayleigh-clear (Mie-cloudy) and GPS-RS winds. The intercepts of Rayleigh-
clear and Mie-cloudy winds are smaller than those for the baseline 2B02. As with the baseline 2B02, no significant
difference was found between the ascending and descending orbits. The biases of Rayleigh-clear and Mie-cloudy winds are
0.45 and –0.71 m s⁻¹, respectively. Both Rayleigh-clear and Mie-cloudy winds have generally met the mission requirements
on systematic errors. These values are almost the same as the intercept of the linear regression line and are smaller than those
for the baseline 2B02. The scaled MAD of Rayleigh-clear winds is 3.97 m s⁻¹ and smaller than that for the baseline 2B02.
On the other hand, the scaled MAD of Mie-cloudy wind is 3.99 m s⁻¹ and almost the same as that for the baseline 2B02.
Martin et al. (2021) estimated the radiosonde representativeness error $\sigma_{r\_GPS-RS}$, and error sources caused by spatial and
temporal displacements need to be considered, in addition to the different measurement geometries of the radiosonde and the
Aeolus observations. They determined that the radiosonde representativeness errors $\sigma_{r\_GPS-RS}$ is 2.48 m s⁻¹ for the Rayleigh-
clear winds, 2.49 m s⁻¹ for the Mie-cloudy winds with 90 km horizontal resolution (corresponding to the baseline 2B02), and
2.66 m s⁻¹ for the Mie-cloudy winds with 10 km horizontal resolution (corresponding to the baseline 2B10). The Aeolus
random error $\sigma_{Aeolus}$ is calculated by using the radiosonde representativeness error $\sigma_{r\_GPS-RS}$,

$$\sigma_{Aeolus} = \sqrt{\sigma_{val}^2 - \sigma_{r\_GPS-RS}^2 - \sigma_{GPS-RS}^2}. \qquad (7)$$

$\sigma_{Aeolus}$ was determined using the Eq. 7 to be 4.01 m s⁻¹ for Rayleigh-clear winds and 3.24 m s⁻¹ for Mie-cloudy winds during
the baseline 2B02 period. During the baseline 2B10 period, $\sigma_{Aeolus}$ was determined to be 3.02 m s⁻¹ for Rayleigh-clear winds,
and 2.89 m s⁻¹ for Mie-cloudy winds. Martin et al. (2021) estimated $\sigma_{Aeolus}$ using the radiosonde observations in the





midlatitudes of the northern hemisphere (23.5–65°N), resulting in $\sigma_{Aeolus}$ of 4.23–4.37 m s$^{-1}$ for Rayleigh-clear winds, and

2.60–2.76 m s$^{-1}$ for Mie-cloudy winds with 90 km horizontal resolution, and 2.97–3.03 m s$^{-1}$ for Mie-cloudy winds with 10

km horizontal resolution. Given that estimates of the representativeness error exhibit large uncertainties (Martin et al., 2021),

the Rayleigh-clear and Mie-cloudy wind random errors during the baseline 2B02 period are consistent with the validation

results of Martin et al. (2021). During the baseline 2B10 period, the Mie-cloudy wind random error is also in good

agreement with the validation result of Martin et al. (2021), whereas the Rayleigh-clear wind random error significantly

decreases. Both Rayleigh-clear and Mie-cloudy wind random errors are close to the mission requirement of 2.5 m s$^{-1}$ in the

free troposphere.


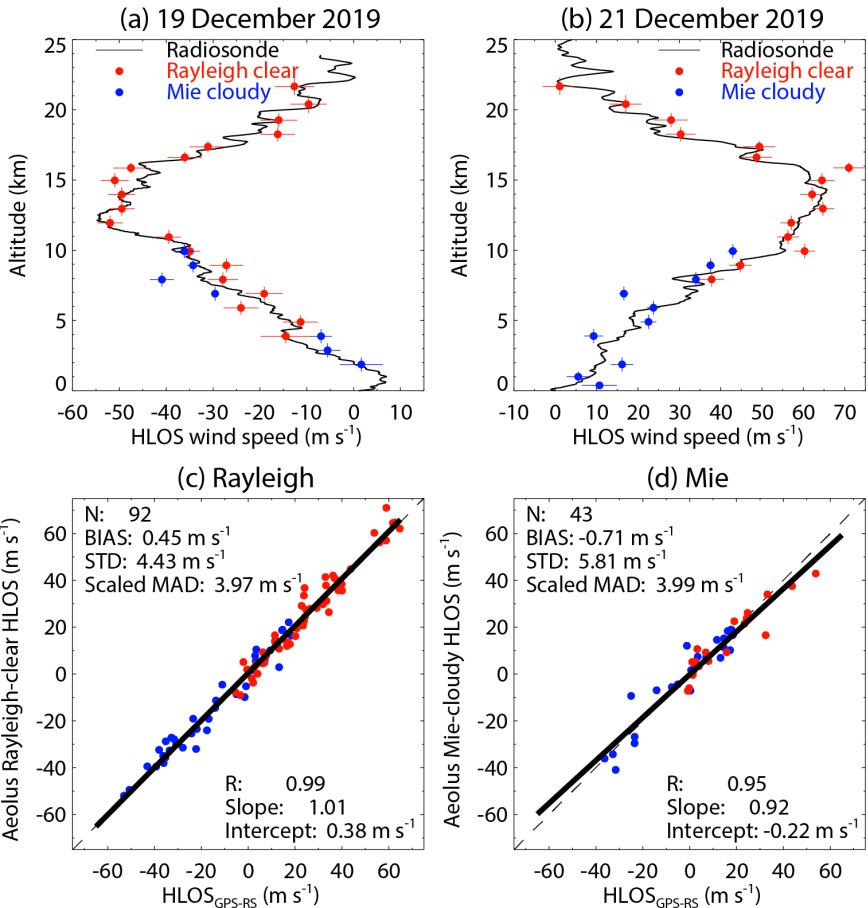

**Figure 14.** (a) HLOS wind speed profiles measured by the radiosonde (thin black line) with the Rayleigh-clear (red) and Mie-cloudy (blue) profiles on 19 December 2019. (b) Same as Fig. 14a but on 21 December 2019. (c, d) Same as Figs. 13c and 13d but for baseline 2B10.



# 6 Summary

We validated the Aeolus L2B data product for Rayleigh-clear and Mie-cloudy winds using operational WPRs, ground-based CDWLs, and GPS-RSs in Japan during the periods of the baseline 2B02 (from 1 October to 18 December 2018) and 2B10 (from 28 June to 31 December 2019 and from 20 April to 8 October 2020). Statistical analyses based on the three independent reference instruments were performed to validate the Rayleigh-clear and Mie-cloudy wind data. In the comparisons of Aeolus and WPR measurements, the vertical distribution of wind difference, the wind bias dependence on latitude and orbit phases, the time series variation of wind differences, and the Rayleigh-clear wind bias dependence on the scattering ratio were investigated.

Statistical analyses of Aeolus HLOS winds and WPR HLOS winds for Rayleigh-clear and Mie-cloudy winds were carried out. For the baseline 2B02, the systematic error was determined to be 1.69 m s⁻¹ for Rayleigh-clear winds and 2.42 m s⁻¹ for Mie-cloudy winds. For the baseline 2B10, the systematic error was determined to be –0.82 m s⁻¹ for Rayleigh-clear winds and –0.51 m s⁻¹ for Mie-cloudy winds. The systematic error for the baseline 2B10 was less than that for the baseline 2B02. For the baseline 2B02, $\sigma_{Aeolus}$ was determined to be 6.71 m s⁻¹ for Rayleigh-clear winds and 5.12 m s⁻¹ for Mie-cloudy winds. For the baseline 2B10, $\sigma_{Aeolus}$ was determined to be 6.42 m s⁻¹ for Rayleigh-clear winds and 4.80 m s⁻¹ for Mie-cloudy winds. The main reason for the large Aeolus random errors is probably related to the large representativeness error due to the large sampling volume of the WPR.

The vertical distributions of differences between Rayleigh-clear or Mie-cloudy winds and WPR winds showed that both Rayleigh-clear and Mie-cloudy biases in all altitude ranges up to 11 km were significantly positive during the baseline 2B02 period. During the baseline 2B10 period, the systematic errors of Rayleigh-clear and Mie-cloudy winds were improved as compared with those during the baseline 2B02 period. The time series of wind speed differences between Aeolus and WPR HLOS winds varied considerably during baseline 2B02 period. Immediately after the launch of Aeolus, both Rayleigh-clear and Mie-cloudy biases were small. With time, the Rayleigh-clear and Mie-cloudy biases increased. Within the baseline 2B02, the Rayleigh-clear and Mie-cloudy biases showed a positive trend. For the baseline 2B10, the biases of Rayleigh-clear and WPR HLOS winds were generally negative at all months except August 2020, but the biases did not show a clear seasonal trend. The biases of Mie-cloudy and WPR HLOS winds gradually fluctuated and did not show a clear seasonal trend. The Rayleigh-clear and Mie-cloudy wind biases were close to 0 m s⁻¹ in 2020. The dependence of the Rayleigh-clear wind bias on the scattering ratio was investigated, showing that the scattering ratio has a minimal effect on the systematic error of the Rayleigh-clear winds during the baseline 2B02 period. On the other hand, during the baseline 2B10 period, there was no significant bias dependence on the scattering ratio.

The statistical analyses based on the ground-based CDWLs at Kobe and Okinawa during the baseline 2B02 and 2B10 periods showed that the agreement between the Aeolus winds and CDWL winds is generally good. For the baseline 2B02, the systematic error was determined to be 0.46 m s⁻¹ (Rayleigh) and 1.63 m s⁻¹ (Mie) at Kobe and 1.08 m s⁻¹ (Rayleigh) and 2.38 m s⁻¹ (Mie) at Okinawa. Except for the Rayleigh-clear winds measured at Kobe, the systematic error did not achieve the





mission requirement. $\sigma_{Aeolus}$ was determined to be 4.49 m s$^{-1}$ (Rayleigh) and 2.93 m s$^{-1}$ (Mie) at Kobe and 5.31 m s$^{-1}$
(Rayleigh) and 3.19 m s$^{-1}$ (Mie) at Okinawa. The Aeolus random errors were larger than those from the validation study
using the airborne 2 μm CDWL (Witschas et al., 2020). The discrepancies were probably caused by the smaller
representativeness error due to the spatial and temporal displacements between Aeolus and airborne CDWL measurements.
For the baseline 2B10, the systematic error was determined to be –0.81 m s$^{-1}$ (Rayleigh) and 0.16 m s$^{-1}$ (Mie) at Kobe and –
0.48 m s$^{-1}$ (Rayleigh) and –0.26 m s$^{-1}$ (Mie) at Okinawa. In contrast to the baseline 2B02, the systematic error significantly
decreased except for the Rayleigh-clear winds measured at Kobe. $\sigma_{Aeolus}$ was determined to be 4.81 m s$^{-1}$ (Rayleigh) and
3.37 m s$^{-1}$ (Mie) at Kobe and 5.21 m s$^{-1}$ (Rayleigh) and 3.30 m s$^{-1}$ (Mie) at Okinawa. In contrast to the comparisons of
Aeolus and WPR measurements, the Aeolus random errors were almost the same as those for the baseline 2B02, and no
improvement of the Aeolus random error was evident.

With the analyses of results obtained from GPS-RSs launched from NICT Okinawa, it was shown that Aeolus can measure
accurately wind profiles with a vertical range up to 25 km and capture the rapid changes in the wind speed profiles such as
the subtropical jet stream. The statistical analyses based on the GPS-RSs also revealed the good performance of Aeolus
during the baseline 2B02 and 2B10 periods. For the baseline 2B02, the systematic error was determined to be 1.00 m s$^{-1}$ for
Rayleigh-clear winds and 2.15 m s$^{-1}$ for Mie-cloudy winds. For the baseline 2B10, the systematic error was determined to be
0.45 m s$^{-1}$ for Rayleigh-clear winds and –0.71 m s$^{-1}$ for Mie-cloudy winds. Both Rayleigh-clear and Mie-cloudy winds
generally met the mission requirements on systematic errors. By taking the radiosonde representativeness error into account,
$\sigma_{Aeolus}$ was determined to be 4.01 m s$^{-1}$ for Rayleigh-clear winds and 3.24 m s$^{-1}$ for the Mie-cloudy winds during the
baseline 2B02 period. During the baseline 2B10 period, $\sigma_{Aeolus}$ was determined to be 3.02 m s$^{-1}$ for Rayleigh-clear winds
and 2.89 m s$^{-1}$ for the Mie-cloudy winds. The random errors of the Rayleigh-clear and Mie-cloudy winds during the baseline
2B02 period were in line with the other validation results. During the baseline 2B10 period, the Aeolus random errors of the
Rayleigh-clear and Mie-cloudy winds were improved as compared with those during the baseline 2B02 period.

**Data availability**

The CDWL and GPS-RS data used in this paper can be provided by the corresponding author (iwai@nict.go.jp) upon request.
The WINDAS data can be downloaded from http://database.rish.kyoto-u.ac.jp/arch/jmadata/data/jma-radar/wprof/original/.
Aeolus data were obtained from the VirES visualization tool (https://aeolus.services/).



**Author contributions**

HI prepared the main part of the paper and performed the statistical analyses of Aeolus data and WPRs, CDWLs, and GPS-RSs data. MA supported the operation of CDWL and GPS-RS measurements. MO performed the GPS-RS measurements. SI
was the principal investigator of validation campaigns in Japan and supported the preparation of this paper and discussed the experimental results. All co-authors helped review the manuscript.

**Competing interests**

The authors declare that they have no conflict of interest.


**Acknowledgements**

We are very grateful to the Japan Meteorological Agency for the operation and maintenance of WINDAS. We are also grateful to Jun Amagai, former Director of NICT Okinawa, for supporting the GPS-RS measurements.

**Financial support**

This work was supported by JSPS KAKENHI Grant Number JP17H06139, JP19K04849, and JP19H01973.

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
