# Peer review of "Validation of Aeolus Level 2B wind products using wind profilers, ground-based Doppler wind lidars, and radiosondes in Japan"

_Atmospheric Measurement Techniques, 2021_

## Referee Comment (RC1)

**General comments:**

The manuscript of Iwai and coauthors reports the comparison of Aeolus HLOS winds over Japan with wind profiler measurements, two ground-based doppler wind lidar and GPS radiosondes. The study is well structured and is found as an important contribution to the Aeolus special edition. The technical set up and statistical methods are comprehensively described. It is appreciated that the systematic and random error estimations are provided for three independent reference data sets and that the results are discussed with respect to other Cal/Val studies. The topic of the submitted paper fits very well to the scientific purposes of the AMT and can be published after addressing some minor comments and suggestions which are listed below.

**Specific comments:**

Abstract:

For an easier readability, I would suggest to round numbers to one decimal place in the abstract

Line 23 – 25: Not well understood. Was the bias during the 2B10 period negative (Rayleigh) and gradually fluctuating (Mie) or was it close to zero? Please try to formulate the message more clearly.

Introduction:

The first paragraph of the introduction needs to be revised. The order of statements of the first four sentences (Line 38 – 42) seems illogical. I suggest to start with the importance of accurate NWP for agriculture, transportation, etc. and then explain the importance of wind field measurements for NWP and furthermore also for climate studies, air quality monitoring, etc. The sentences in Line 43 –44 are kind of misleading. Are you here referring to the entire global observing network or do you mean, that measurements such as radiosondes, wind profilers (WPRs), ground-based Doppler wind lidars (DWLs), and aircrafts provide accurate vertical profiles and that they are limited from the global perspective? If the latter is the case you have to distinguish between single level aircraft measurements at flight level and aircraft ascents. In the following part I suggest to make the limitations of wind measurements from satellite radiances and AMVs more clearly. I miss the information that satellite radiances only provide information about the mass field which leads to particularly strong restriction in the absence of geostrophic balance. The limited accuracy of AMV single-level winds is mainly caused by significant systematic and correlated errors due to uncertainties of their height assignment (see., Folger and Weissmann, 2014 or Bormann et al., 2003).

Line 50 - 51: Please be precises - Aeolus is the name of the earth explorer mission/satellite and not the name of the DWL. Regarding the following sentence this can be confusing.

Line 57 – 59: For me the expression 'and so forth' implies that further purposes are obviously. Is this really the case? Furthermore, I suggest to put some reference here (e.g. ESA, 1999: The four candidate Earth Explorer core missions - Atmospheric Dynamics Mission or ESA, 2001: ADM-Aeolus Mission Requirements Document)

Since some phrases are very similar to Belova et al. 2021, I suggest to add this study to your reference

Section 3:

I suggest to already mention that you use the vertical range bins of Aeolus as vertical collocation criteria in the paragraphs about the horizontal and temporal collocation criteria when comparing the Aeolus winds to WPRs, CDWLs and RSs. This is a question the reader may asks himself at this point.

In section 2 about the Aeolus data and in the abstract, you mention the three validation periods, whereby you write that the first period is from 1 October to 18 December. In Section 3 you write that you are using the WPRs and CDWLs for the comparisons from 18 October 2018 to 15 and 11 May 2019. In table 4 you also have the 20 December 2018 as validation day for comparisons with radiosondes. So, you also used the Aeolus data with baseline 2B02 till May 2019? Please correct for this consistently in the whole paper.

Section 4:

Line 196: Please shortly describe how the Aeolus azimuth angle is defined.

Line 213: Maybe add some additional validation studies like Lux et al., 2020 or Martin et al.,2021 as reference.

Section 5:

Numbers are sometimes rounded to two decimal places, sometimes to only one decimal places. As your plots and tables are showing the statistical results which are discussed in section 5 as numbers rounded to two decimal places, I suggest to also do so in the text

Line 261: Actually, you cannot compare your estimates of systematic and random errors for HLOS winds with the values of Lux et al. 2020.  The A2D measures LOS with off-nadir of 20° so that it is not possible to calculate HLOS (in Aeolus direction). Furthermore, the azimuth directions are not identically. The paper concentrates on a comparison in LOS space which is not equivalent to comparisons in HLOS space. You have to multiply it with a factor of 1/sin(37), see section 4.2 of Lux et al. 2020.

In the first part of your paper (section 2), you stress that you are investigating three different validation periods. However, in section 5.1.1 and 5.1.2 the reader my wonders why you are only talking about two, the 2B02 and 2B10, periods. Aren't there any differences in the statistics between the reprocessed data set and the data from April 2020 on? Shortly explain why you are combing the two time periods, refer to section 5.1.3 where you are showing the time series or consider to just talk about two validation periods in section 2 – the 2B02 and the 2B10 period while the 2B10 period is composed of the M1 bias corrected observations and the reprocessed data set.

In the end of section 5.1.1 you could point out that the reduced bias of the 2B10 period compared to 2B02 is most likely due to the M1 bias correction.

Line 313: There is a difference in bias for Rayleigh-clear 2B10 between ascending and descending of 0.6 m/s. Is this really not significant?

Figure 6 and 7: A second axis with the number of compared data points could be nice. But this is only a suggestion.

In section 5 you are sometimes disarranging tenses. For example, line 329 and line 330 should be in present like in the sentences before and after. Please look through your paper once again carefully with focus on tenses.

Line 332: This statement kind of confuses the reader at this point. Improved compared to what?

Section 5.1.2, Figure 7: I find it prominent that the bias for is tending to get more negative with height, while for ascending it is kind of the other way around. The negative trend with height for the descending HLOS winds is also visible in Figure 6. Maybe consider to mention this.

Line 349: Towards the end of 2B02 period (after April 2019) it looks more like a negative trend - especially for descending orbit.

Line 353: I suggest to add a reference about the laser performance here:

> Reitebuch et al., (2020b), Assessment of Aeolus performance and bias correction results from the Aeolus DISC, Aeolus Cal/Val and Science Workshop 2020, https://nikal.eventsair.com/QuickEventWebsitePortal/2nd-aeolus-post-launch-calval-and-science-workshop/aeolus

> Reitebuch, Oliver, Christian Lemmerz, Oliver Lux, Uwe Marksteiner, Stephan Rahm, Fabian Weiler, Benjamin Witschas, et al. (2020). "Initial Assessment of the Performance of the First Wind Lidar in Space on Aeolus". Edited by D. Liu, Y. Wang, Y. Wu, B. Gross, and F. Moshary. EPJ Web of Conferences 237: 01010. https://doi.org/10.1051/epjconf/202023701010

> Also consider, that changes in the Aeolus range bin settings can lead to changes in the random error.

Line 354: It is the same for the descending orbit. Furthermore, you already mentioned the overall negative bias for Rayleigh winds in line 351. So, maybe consider to just remove this sentence.

Line 358: In fact, the Mie signal does depend on the laser energy, I think. But it depends also on the presence of aerosols or hydrometeors (see Martin et al. 2020 end of section 3.1.). That's why the effect is not visible in the random error that prominent compared to the Rayleigh winds. Please correct me if I am wrong.

Line 360: Approached zero towards September 2020, in my view.

Line 370: I don't really see an increase in Rayleigh bias with larger scattering ration. Isn't the bias more fluctuating and almost the same for scatstering ration 1.1 and 1.4?

For the profiles of the comparisons of Aeolus with radiosondes (Fig. 13 and 14 (a),(b)), it would be nice to have the information of orbit phase in the caption.

Line 457: In my opinion the Mie bias is not almost the same as the Mie bias obtained by Baars et al 2020. Its 1.2 m/s larger. Probably due to small number of radiosonde launches of Baars et al, different location and meteorological conditions or different distance between the measurements?

Line 477 – 479: The sentence is not completely right in my opinion. Martin et al 2021 estimated the representativeness error by considering different error sources, not the representativeness error and error sources. In your case the representativeness errors result from the different measurement geometries and from the collocation criteria (spatial and temporal displacement).

Since you use the scaled MAD as estimate for the random error in your paper, I suggest to name the sigma Aeolus 'Aeolus observational error' instead of 'Aeolus random error'. Otherwise, it could be misleading.

Summary

Line 502: You say that you investigated the bias dependence on latitude, but there is no such part in the manuscript.

Line 521: Again, in my opinion the bias only is getting close to zero towards September 2020. In the early summer months 2020 it is as close to zero than in 2019.

I miss a nice closing sentence to round off the whole study. You could mention that despite you used three independent data sets with quite different sample sizes the differences in bias estimation are not greater than 1 m/s (except of Rayleigh 2B10). What's your conclusion of the good concordance between the three independent validation data sets? What's the importance of this study for the Aeolus mission and the use of the Aeolus HLOS winds in NWP? You can also give kind of an outlook or suggestions for improvements of your study.

**Technical corrections:**

Line 9: check spelling 'onboard'

Line 16 - 18: Avoid duplication. Two sentences with the same message. I would suggest the following wording: "The statistical comparisons for the baseline 2B10 period show smaller biases, –0.8 – 0.5 m s−1 for the Ryleigh clear and –0.7 – 0.2 m s−1 for the Mie cloudy winds."

Line 57: please correct 'purposes'

Line 60: please correct 'requires'

Line 101 – 106: for better understanding consider to rephrase like this: "In this study, we used three different periods during the processor baseline 2B02 and 2B10 periods to assess L2B data products: 1 October to 18 December 2018 (2B02), 28 June to 31 December 2019 (2B10) and 20 April to 8 October 2020 (2B10). The first period with baseline 2B02 was within the commissioning phase. The L2B data products with the 2B10 baseline include a bias correction for ALADIN's telescope primary (M1) mirror temperature variation (Rennie and Isaksen, 2020; Rennie et al. 2021*) and have been available for new observations since April 2020. The L2B winds from 28 June to 31 December 2019 are a homogeneous reprocessed dataset using also the 2B10 processor version." *I suggest to add the reference Rennie et al. 2021 for more information about the M1 bias correction.

Line 126: please add a space between number and unit (294 m)

Line 198: it seems that the spacing between the equation environment and the text is missing

Line 205: consider to rephrase as follows: "...estimated HLOS errors are 2.3 m/s during both, baseline 2B01 and 2B10."

Line 228: it seems that the spacing between the equation environment and the text is missing

Line 267 and 292: consider to remove the term 'large' to not sound judgmental

Line 294: almost same sentence than Line 271. I suggest to rephrase as follows: "Again, the discrepancies may be caused by ... "

Line 356: wrong figure reference - Fig. 9d not Fig 9c

Line 440: please modify: "...the reason for that.."

Line 440 – 441: consider to rephrase: "Potentially, large horizontal wind gradients in this height region have an influence on the differences.

Line 482: consider to rephrase; "The Aeolus observation error considering the representativeness error in addition to the radiosonde observational error can be calculated as follows:..."

Line 503 and 511: I think there is no need for a paragraph here. Rather, I suggest to put a paragraph before you start to summarize the WPR validation in line 500, like you also did before the CDWL validation and before the radiosonde validation summary. That seems more conclusive for me.

Line 504 – 505: could remove this sentence. It's kind of repetition of the sentence in line 499 – 500.

Line 513: be careful with using the expression significant. Actually, this should be based on a statistical significance test. Better remove 'significantly' here.

Line 518 – 519: this does not make sense. Either "...the bias of Rayleigh-clear HLOS winds were generally negative..." or "...the systematic differences between the Rayleigh clear and WPR HLOS winds were generally negative..."

Line 520: consider to modify: "... and did also not show a clear seasonal trend."

Line 533: next line before the minus symbol

Line 534: again, maybe reconsider the use of significantly here

---

## Referee Comment (RC2)

Review of the article

**"Validation of Aeolus Level 2B wind products using wind profilers, ground-based Doppler wind lidars, and radiosondes in Japan"**

submitted by H. Iwai et al.
**(AMT)**

**Scientific significance: Excellent**

The paper deals with validation of the Aeolus HLOS winds based on three different reference instruments (Windprofiler, Doppler wind lidars and radiosondes) during different time periods of the mission covering the two different Aeolus laser transmitters (FM-A and FM-B) as well as different data processor versions (2B02 and 2B10). The study is well structured and is an important contribution to the global Aeolus cal/val activities and fits especially well the Aeolus AMT special issue.

**Scientific quality: Good**

The paper addresses all information that are needed to understand the quality of the used reference measurements data as well as the methods that are used for comparison. Minor issues that could even improve the presented results are addressed below.

**Presentation quality: Good**

The paper manuscript is clearly structured, all used methodologies are well explained and all figures are clearly visible. The text is well and clearly written. Only minor points are mentioned below that may help to further improve the manuscript.

**Review Summary**

The paper manuscript "Validation of Aeolus Level 2B wind products using wind profilers, ground-based Doppler wind lidars, and radiosondes in Japan" H. Iwai et al. deals with the validation of Aeolus horizontal line-of-sight (HLOS) winds during different time periods (Oct to Dec 2018, Jun to Dec 2019 and Apr to Oct 2020) based on three different reference data sources coming from windprofilers, two coherent Doppler wind lidars as well as from radiosondes launched from Japan. Hence, these comparisons validate data obtained with the two different Aeolus laser transmitters FM-A and FM-B as well as data based on two different processor versions 2B02 and 2B10. The paper manuscript is clearly structured, all used methodologies are well explained and all figures are clearly visible. The content of the paper is well suited for AMT and especially for the Aeolus special issue. It is recommended to accept the paper manuscript after addressing the minor points that are mentioned below.

**Detailed comments**

- **General:**
    - o The entire analysis uses Aeolus data that is processed with two different processor versions 2B02 and 2B10. Could you somehow address the main differences that were implemented in the processor between version 2B02 and 2B10. This would help the reader to understand why one would expect really differences especially in the systematic error due to the implementation of the M1 mirror temperature correction or the hot pixel correction that was implemented.
    - o You compare your determined results with different studies (GPS-RS results with Martin et al.; CDWL results with Witschas et al., WPR results with Guo et al.). But actually, you can use the data of all three instruments as reference. Thus, I am missing a detailed comparison of your own results from the different measurement data of the WPR, the CDWL and the GPS-RS. This should also be available in the summary, maybe with a table that summarizes to retrieved systematic errors based on the respective reference data sets.
    - o Sometimes there is a confusion with the "-" sign which sometimes denotes a minus and sometimes denotes a "to". Thus, it is suggested to replace the "-" sign with the word "to" and only use "-" as a minus sign.
    - o If possible, I would not spread units over two lines.

- **Fig.1, caption:** bule → blue

- **Line 92:** "two interferometers"…actually, as you also write later in the text, there are three interferometers: one Fizeau interferometer, and two Fabry-Perot interferometers all of them illuminated sequentially.

- **Line 98:** "Fizeau and Fabry Perot as narrowband filters" Somehow this is true, but the bandwidth is still different by a factor of 5 (Fizeau interferometer about 2 GHz, Fabry-Perot interferometers about 11 GHz).

- **Line 99:** "After passing through some optics". You could skip that part of the sentence as it provides no information.

- **Line 106:** "Commissioning phase" which was from x.x to x.x?.

- **Line 110:** "Several different technical, instrumental, and retrieving checks account for this flag." → Could you give a few examples which checks are performed and considered for deriving the validity flag?

- **Line 126:** "There is no significant difference between wind profiler winds and radiosonde winds in the biases and root mean square errors. → What do you mean with "significant"? Is there a difference? If yes, it would be better to quantify here. How do you determine the bias of WPR measurements or rather radiosondes?

- **Line 126:** "294m" → 294 m (space)

- **Line 148:** "Doppler beam swinging (DBS) technique ". Is this a well-known technique? Could you add a reference here?

- **Line 149:** "The Doppler velocity spectra for all range bins of each beam were obtained 10,000 times on average. Since the PRF was 10 kHz, the accumulation time of each beam was 10 s". If I understand it correctly, the averaging time is one second, or you accumulate 100000 spectra. Or do I misunderstand something here?

- **Line 152:** "Maximum likelihood estimator" Can you give a reference here?

- **Line 153:** "Bias was estimated at 0.02 m s–1 using measurements from a stationary hard target." → I guess, this value is only true for single LOS measurements, isn't it? If yes, this should be clarified here.

- **Line 159:** "As mentioned earlier, we averaged Doppler velocity spectra for all range bins of each beam from 30 min before to 30 min after the passage of Aeolus, and then the vertical profiles of horizontal wind speed and wind direction were acquired by the DBS technique." → Is it true that you averaged all spectra for 60 minutes and then you determine the mean wind speed? Using such an approach, I would expect the center peak in your power spectrum to be rather broad. Wouldn't it be better to calculate the wind speed on a e.g. one-minute average, and then calculate the mean over the 60 data points? Furthermore, I do not understand why you calculate and average for your CDWL comparison. Couldn't you actually just use the profile measured directly during the Aeolus overpass?

- **Line 229:** "Note that this estimation of $\sigma$ includes the representativeness error due to the spatial and temporal mismatch between Aeolus and reference instruments' measurements." → The representativeness error is likely to be different for the different reference instrument measurements. For instance, for the CDWL measurements you could decrease the representativeness error by decreasing the temporal averaging time to only a few minutes. Have you tried if this changes the random error for the comparison?

- **Line 248:** "Furthermore, the range-gate settings of Aeolus were changed on 26 February 2019, which also increased the number of available data points during the baseline 2B10 period." → What was changed in particular? More range gates in the troposphere, less in the stratosphere? Or just increasing the range bin size? Here it would be helpful to get more details.

- **Line 252:** "slightly positive". As the bias is 1.6 to 1.8 m/s which is a factor of more than two larger the originally specified, I would skip the word "slightly" here.

- **Line 261:** "Comparison to A2D data". → You should be careful when comparing to A2D data as they compare LOS winds and not HLOS winds.

- **Line 289:** It is worth mentioning here that only ascending orbits were underflown during the WindVal III campaign.

- **Line 311:** "The main reason for not yet achieving the mission requirement for random errors is probably related to the large representativeness error due to the large sampling volume of the WPR." → Also the Aeolus laser pulse energy is remarkably less than specified. With the representativeness error you would argue that the actual random error of Aeolus L2B data could meet the requirements. However, this is not true.

- **Line 320:** "…there are relatively many paired data 320 points for comparison (Fig. 6a)." → What does relatively many mean? Can you please quantify. It would also be very helpful to have this information plotted in Fig.6/Fig.7. This would give the possibility to understand how likely the shown bias trend is. For instance, is the negative Mie bias for descending orbits in 10 km altitude real, or just a result of very few data points and thus not reliable.

- **Line328:** "The systematic error was less than that of the baseline 2B02." → Why? Here, you could refer to the differences in 2B02 and 2B10 which I suggested to discuss in the previous part of the manuscript.

- **Line 332:** "However, this result is different from that in the other validation studies conducted during the baseline 2B10 period (Guo et al., 2021)." → What is different? What is the result by Guo et al.? Would be good to write one sentence here such that this information is available without reading Guo et al., 2021.

- **Line 339:** "(descending) orbit, the minimum (maximum) bias is −1.93 (0.54) m s−1 in the altitude range of 5–6 (4–5) km." → this is confusing. Is the bias of -1.93 m/s corresponding to ascending or descending orbits?

- **Fig. 8:** "Monthly averages" → Why do you calculate monthly averages and do not show a time series on a daily basis?

- **Line 358:** "because the Mie return signal does not depend on the laser energy (Martin et al., 2021)." → I would not write it that strong. Indeed, compared to Rayleigh returns, Mie signals are much more depending on the atmospheric backscatter. But of course, if the laser pulse energy is too low, one would not be able to measure at all.

- **Line 369:** "During the baseline 2B02 period, the bias of Rayleigh-clear and WPR HLOS winds slightly increased as the scattering ratio increased (Fig. 10a)." → Would this be better visible when calculating daily means instead of monthly means?

- **Line 391:** "result is similar to that in the comparisons of Aeolus and WPR measurements." → which provides biases of x.x m/s. Would be good to repeat the numbers here.

- **Line 394:** "The values are smaller than the scaled MADs of Rayleigh-clear (Mie-cloudy) versus WPR winds." → Why do you think is this the case? The representativeness is similar, and the respective measurements errors of the WPR or rather the CDWL is corrected, isn't it?

- **Line 423:** "The Rayleigh-clear winds show good coverage and closely follow the shape of the wind profile at altitudes higher than 2 km." → Any ideas or hints what causes outliers as e.g. the one at 8 km for the Rayleigh-clear winds? Still aerosol contamination, as the range gates below provides a valid Mie wind? Would be interesting to see if there is an issue with the cross-talk correction of Mie signals in the Rayleigh channel...

- **Line 435:** "…the clouds were partly existent in the Aeolus observational domain" → This means that clouds are not sufficiently corrected or filtered out in the Rayleigh data product? E.g., the Rayleigh-clear wind in between 3-4 km where also clouds were partly present shows a quite large bias. Is there any way to analyze this altitude in more detail, e.g., by analyzing single Aeolus "measurements" instead of "observations", or is this information not contained in the L2B data product?

- **Line 440:** "There is a possibility regarding horizontal wind gradients in this height region." → You have actual wind speed and direction available from the radiosonde, right? If yes, it might be good to show if there was indeed a wind shear in this altitude.

- **Line 477:** "Martin et al. (2021) estimated the radiosonde representativeness error, and error sources caused by spatial and temporal displacements need to be considered," → How do they do that? Based on comparison to measurements or with respect to theoretical assumptions? This would be an interesting side note.

---

## Author Comment (AC1)

Title: Validation of Aeolus Level 2B wind products using wind profilers, ground-based Doppler wind lidars, and radiosondes in Japan

Author(s): Hironori Iwai et al.

MS No.: amt-2021-243

MS type: Research article

Special Issue: Aeolus data and their application (AMT/ACP/WCD inter-journal SI)

**Responses to referee #1**

First, we would like to thank the referee #1 for your constructive comments that helped us improve our manuscript. In the present document, we provide our responses to the comments. The comments of the referee are reported in black font, our responses and the corresponding modifications in the manuscript in blue font, and the changes to the original manuscript in red font.

**General comments:**

The manuscript of Iwai and coauthors reports the comparison of Aeolus HLOS winds over Japan with wind profiler measurements, two ground-based doppler wind lidar and GPS radiosondes. The study is well structured and is found as an important contribution to the Aeolus special edition. The technical set up and statistical methods are comprehensively described. It is appreciated that the systematic and random error estimations are provided for three independent reference data sets and that the results are discussed with respect to other Cal/Val studies. The topic of the submitted paper fits very well to the scientific purposes of the AMT and can be published after addressing some minor comments and suggestions which are listed below.

We are grateful for the referee's appropriate and positive comments on our manuscript. We revised the manuscript according to your comments and the details are shown as follows.

**Specific comments:**

Abstract:

For an easier readability, I would suggest to round numbers to one decimal place in the abstract

We accepted your suggestion and rounded numbers to one decimal place in the Abstract.

Line 23 – 25: Not well understood. Was the bias during the 2B10 period negative (Rayleigh) and gradually fluctuating (Mie) or was it close to zero? Please try to formulate the message more clearly.

Thanks for the suggestion. We changed the sentences for further clarification:

For the baseline 2B10, the Rayleigh-clear wind bias was generally negative at all months except August 2020, and Mie-cloudy wind bias gradually fluctuated. Both Rayleigh-clear and Mie-cloudy biases did not show a marked seasonal trend and approached zero towards September 2020.

Introduction:

The first paragraph of the introduction needs to be revised. The order of statements of the first four sentences (Line 38 – 42) seems illogical. I suggest to start with the importance of accurate NWP for agriculture, transportation, etc. and then explain the importance of wind field measurements for NWP and furthermore also for climate studies, air quality monitoring, etc. The sentences in Line 43 –44 are kind of misleading. Are you here referring to the entire global observing network or do you mean, that measurements such as radiosondes, wind profilers (WPRs), ground-based Doppler wind lidars (DWLs), and aircrafts provide accurate vertical profiles and that they are limited from the global perspective? If the latter is the case you have to distinguish between single level aircraft measurements at flight level and aircraft ascents. In the following part I suggest to make the limitations of wind measurements from satellite radiances and AMVs more clearly. I miss the information that satellite radiances only provide information about the mass field which leads to particularly strong restriction in the absence of geostrophic balance. The limited accuracy of AMV single-level winds is mainly caused by significant systematic and correlated errors due to uncertainties of their height assignment (see., Folger and Weissmann, 2014 or Bormann et al., 2003).

Thanks for the suggestion. The first paragraph of Sect. 1 was changed to:

Accurate numerical weather prediction (NWP) is useful for commercial activities such as agriculture, fisheries, construction, transportation, and energy development, and for daily life. Since wind is one of the fundamental meteorological variables describing the atmospheric state, it is very important to understand the evolution and structure of winds for NWP. Measurement of the three-dimensional global wind field is crucial for NWP and furthermore also for air quality monitoring and forecasting, climate studies, and various meteorological studies. The wind observations obtained by the global meteorological observing system, which contains radiosondes, wind profilers (WPRs), and aircrafts, are routinely assimilated in NWP models. The radiosondes, WPRs, and aircrafts during takeoff and landing provide accurate and precise vertical wind profiles. However, the observational coverage is limited from the global perspective. Satellite-borne microwave scatterometers and radiometers can estimate ocean surface vector winds using microwave return from the ocean roughness. Although these instruments well capture mesoscale wind field at the ocean surface, they do not provide any profiling information. Atmospheric motion vectors (AMVs) can be retrieved from cloud and water vapour motions derived from geostationary and polar-orbit satellite images (e.g., Bormann et al., 2003). AMVs have a large coverage area and high temporal and horizontal resolutions, but the limited accuracy of AMV winds is mainly caused by significant systematic and correlated errors due to uncertainties of their height assignment (e.g., Folger and Weissmann, 2014).

Line 50 - 51: Please be precises - Aeolus is the name of the earth explorer mission/satellite and not the name of the DWL. Regarding the following sentence this can be confusing.

We changed the sentence as follows:

The European Space Agency (ESA) launched on 22 August 2018 the first space-based DWL on board the satellite Aeolus, for obtaining global wind profiles (Kanitz et al., 2019; Reitebuch et al., 2020a).

Line 57 – 59: For me the expression 'and so forth' implies that further purposes are obviously. Is this really the case? Furthermore, I suggest to put some reference here (e.g. ESA, 1999: The four candidate Earth Explorer core missions

- Atmospheric Dynamics Mission or ESA, 2001: ADM-Aeolus Mission Requirements Document)

Thanks for the suggestion. We removed the phrase 'and so forth' and put two references (ESA, 1999; Ingmann and Straume, 2016) here.

Since some phrases are very similar to Belova et al. 2021, I suggest to add this study to your reference

We added Belova et al. (2021) to the reference.

Section 3:

I suggest to already mention that you use the vertical range bins of Aeolus as vertical collocation criteria in the paragraphs about the horizontal and temporal collocation criteria when comparing the Aeolus winds to WPRs, CDWLs and RSs. This is a question the reader may asks himself at this point.

Thanks for the suggestion. We removed the sentence about the vertical collocation criteria in Sect. 4 and added the sentences about the vertical collocation criteria when comparing the Aeolus winds to WPRs, CDWLs and RSs to Section 3.1, 3.2, and 3.3, respectively.

Section 3.1:

There is also a difference in the vertical resolution between Aeolus measurements and WPR measurements. The horizontal wind speed and wind direction measured by the WPRs were averaged to the Aeolus bin by using the top and bottom altitudes given in the Aeolus L2B data product.

Section 3.2:

As with the WPR, the horizontal wind speed and wind direction measured by the CDWLs were averaged to the Aeolus bin.

Section 3.3:

As with the WPR, the horizontal wind speed and wind direction measured by the GPS-RSs were averaged to the Aeolus bin.

In section 2 about the Aeolus data and in the abstract, you mention the three validation periods, whereby you write that the first period is from 1 October to 18 December. In Section 3 you write that you are using the WPRs and CDWLs for the comparisons from 18 October 2018 to 15 and 11 May 2019. In table 4 you also have the 20 December 2018 as validation day for comparisons with radiosondes. So, you also used the Aeolus data with baseline 2B02 till May 2019? Please correct for this consistently in the whole paper.

We corrected the period of baseline 2B02 in Sect. 2 as follows:

In this study, we used three different periods during the processor baseline 2B02 and 2B10 periods to assess L2B data products: 1 October 2018 to 15 May 2019 (2B02), 28 June to 31 December 2019 (2B10) and 20 April to 8 October 2020 (2B10).

Section 4:

Line 196: Please shortly describe how the Aeolus azimuth angle is defined.

We added the phrase as follows:

which is obtained from L2B data product

Line 213: Maybe add some additional validation studies like Lux et al., 2020 or Martin et al.,2021 as reference.

We added Belova et al. (2021), Lux et al. (2020), and Martin et al. (2021) as reference.

Section 5:

Numbers are sometimes rounded to two decimal places, sometimes to only one decimal places. As your plots and tables are showing the statistical results which are discussed in section 5 as numbers rounded to two decimal places, I suggest to also do so in the text

We rounded all numbers to two decimal places in Sect. 5. For example:

between 1.6 and 1.8 m s$^{-1}$ -> between 1.63 and 1.76 m s$^{-1}$

Line 261: Actually, you cannot compare your estimates of systematic and random errors for HLOS winds with the values of Lux et al. 2020. The A2D measures LOS with off-nadir of 20° so that it is not possible to calculate HLOS (in Aeolus direction). Furthermore, the azimuth directions are not identically. The paper concentrates on a comparison in LOS space which is not equivalent to comparisons in HLOS space. You have to multiply it with a factor of 1/sin(37), see section 4.2 of Lux et al. 2020.

Thanks for the suggestion. We modified the sentences as follows:

Lux et al. (2020b) compared the Rayleigh-clear winds measured along the Aeolus LOS with LOS winds measured with the ALADIN Airborne Demonstrator (A2D) during the WindVal III validation campaign carried out in central Europe from 17 November to 5 December 2018 (i.e., during the baseline 2B02 period). They reported a bias of 2.56 m s$^{-1}$ with a scaled MAD of 3.57 m s$^{-1}$, corresponding to HLOS values of 4.25 and 5.93 m s$^{-1}$, respectively.

And we removed the sentence "They also reported that the slope of the linear regression line and the correlation coefficient of Rayleigh-clear versus A2D winds were 0.83 and 0.80, respectively."

In the first part of your paper (section 2), you stress that you are investigating three different validation periods. However, in section 5.1.1 and 5.1.2 the reader my wonders why you are only talking about two, the 2B02 and 2B10, periods. Aren't there any differences in the statistics between the reprocessed data set and the data from April 2020 on? Shortly explain why you are combing the two time periods, refer to section 5.1.3 where you are showing the time series or consider to just talk about two validation periods in section 2 – the 2B02 and the 2B10 period while the 2B10 period is composed of the M1 bias corrected observations and the reprocessed data set.

Thanks for the suggestion. We mentioned two validation periods in Sect. 2 as follows:

We mainly discuss the measurement performance of Aeolus for Rayleigh-clear and Mie-cloudy winds during the baseline 2B02 and 2B10 periods. The baseline 2B10 period is composed of the M1 mirror and hot pixel bias corrected observations and the reprocessed data set.

In the end of section 5.1.1 you could point out that the reduced bias of the 2B10 period compared to 2B02 is most likely due to the M1 bias correction.

Thanks for the suggestion. We added the sentence as follows:

The reduced bias of the baseline 2B10 period compared to the baseline 2B02 is most likely due to the M1 mirror bias correction (Rennie and Isaksen, 2020; Weiler et al. 2021b) and the improvement of the hot pixel correction.

Line 313: There is a difference in bias for Rayleigh-clear 2B10 between ascending and descending of 0.6 m/s. Is this really not significant?

Thanks for the suggestion. We modified the sentence as follows:

Although, from the statistical comparisons, there is no significant difference between the ascending and descending orbits with respect to the Rayleigh-clear and Mie-cloudy winds during the baseline 2B02 period, the absolute biases of the Rayleigh-clear and Mie-cloudy winds are slightly larger for the ascending orbit than for the descending orbit during the baseline 2B10 period.

Figure 6 and 7: A second axis with the number of compared data points could be nice. But this is only a suggestion.

Thanks for the suggestion. Since it is difficult to add a second axis with the number of compared data points, we plotted the vertical profile of the number of compared data points as shown in Figs. S1 and S2.

[Figure]

**Figure S1.** Vertical profiles in 1 km bins of the number of compared data points between the Aeolus and WPR HLOS winds for (a, b, c) Rayleigh-clear winds and (d, e, f) Mie-cloudy winds for (a, d) all data and (b, e) ascending and (c, f) descending orbits for baseline 2B02.

[Figure]

**Figure S2.** Same as Fig. S1 but for baseline 2B10.

In section 5 you are sometimes disarranging tenses. For example, line 329 and line 330 should be in present like in the sentences before and after. Please look through your paper once again carefully with focus on tenses.

We corrected the tenses in the whole paper.

Line 332: This statement kind of confuses the reader at this point. Improved compared to what?

We removed this sentence as suggested.

Section 5.1.2, Figure 7: I find it prominent that the bias for is tending to get more negative with height, while for ascending it is kind of the other way around. The negative trend with height for the descending HLOS winds is also visible in Figure 6. Maybe consider to mention this.

Thanks for the suggestion. We add the sentences in Sect. 5.1.2 as follows:

Although there are some local maxima and minima, Rayleigh-clear biases tend to get more negative with altitude above 2 km altitude.

As with the baseline 2B02, both Rayleigh-clear and Mie-cloudy biases show a negative trend with altitude for all data and descending orbit, whereas they show a positive trend for ascending orbit.

Line 349: Towards the end of 2B02 period (after April 2019) it looks more like a negative trend - especially for descending orbit.

We modified this sentence as follows:

The Rayleigh-clear and Mie-cloudy biases tend to get more positive until April 2019, whereas they show a negative trend at the end of the baseline 2B02 period.

We modified the related sentence in Sect. 6 as follows:

Within the baseline 2B02, the Rayleigh-clear and Mie-cloudy biases showed a positive trend until April 2019, whereas they show a negative trend at the end of the baseline 2B02 period.

Line 353: I suggest to add a reference about the laser performance here:

Reitebuch et al., (2020b), Assessment of Aeolus performance and bias correction results from the Aeolus DISC, Aeolus Cal/Val and Science Workshop 2020, https://nikal.eventsair.com/QuickEventWebsitePortal/2nd-aeolus-post-launch-calval-and-science-workshop/aeolus

Reitebuch, Oliver, Christian Lemmerz, Oliver Lux, Uwe Marksteiner, Stephan Rahm, Fabian Weiler, Benjamin Witschas, et al. (2020). "Initial Assessment of the Performance of the First Wind Lidar in Space on Aeolus". Edited by D. Liu, Y. Wang, Y. Wu, B. Gross, and F. Moshary. EPJ Web of Conferences 237: 01010. https://doi.org/10.1051/epjconf/202023701010

We added Reitebuch et al. (2020a and 2020b) about the laser performance here and to the reference.

Also consider, that changes in the Aeolus range bin settings can lead to changes in the random error.

Thanks for the suggestion. We added the sentence as follows:

The higher range-bin resolution in the lower troposphere after 21 October 2019 can also lead to increase in the random error.

Line 354: It is the same for the descending orbit. Furthermore, you already mentioned the overall negative bias for Rayleigh winds in line 351. So, maybe consider to just remove this sentence.

We removed the sentence as suggested.

Line 358: In fact, the Mie signal does depend on the laser energy, I think. But it depends also on the presence of aerosols or hydrometeors (see Martin et al. 2020 end of section 3.1.). That's why the effect is not visible in the random error that prominent compared to the Rayleigh winds. Please correct me if I am wrong.

You are right. We corrected the sentence as follows:

There is no significant increase in the standard deviations of Mie-cloudy winds with time, because the Mie return signal does not only depend on the laser energy but also on the presence of aerosols or clouds (Martin et al., 2021).

Line 360: Approached zero towards September 2020, in my view.

Thanks for the suggestion. We modified the sentence as follows:

However, the biases for both orbit phases approach zero towards September 2020.

Line 370: I don't really see an increase in Rayleigh bias with larger scattering ration. Isn't the bias more fluctuating

and almost the same for scattering ration 1.1 and 1.4?

You are right. We modified and added the sentences as follows:

During the baseline 2B02 period, the biases of Rayleigh-clear and WPR HLOS winds were positive in the range of 1.38 and 2.21 m s$^{-1}$ (Fig. 10a). Since there was no significant bias dependence on the scattering ratio, the influence of the cross talk of narrowband Mie return signals to the Rayleigh channel was not confirmed. This result is different from that obtained in Witschas et al. (2020).

We modified the related sentence in Abstract as follows:

The dependence of the Rayleigh-clear wind bias on the scattering ratio was investigated, showing that there was no significant bias dependence on the scattering ratio during the baseline 2B02 and 2B10 periods.

We modified the related sentences in Sect. 6 as follows:

The dependence of the Rayleigh-clear wind bias on the scattering ratio was investigated, showing that the influence of the cross talk of Mie signals to the Rayleigh channel was not confirmed during the baseline 2B02 period. As with the baseline 2B02, there was no significant bias dependence on the scattering ratio during the baseline 2B10 period.

For the profiles of the comparisons of Aeolus with radiosondes (Fig. 13 and 14 (a),(b)), it would be nice to have the information of orbit phase in the caption.

Thanks for the suggestion. We added the information of orbit phase in the caption of Figs. 13 and 14.

Line 457: In my opinion the Mie bias is not almost the same as the Mie bias obtained by Baars et al 2020. Its 1.2 m/s larger. Probably due to small number of radiosonde launches of Baars et al, different location and meteorological conditions or different distance between the measurements?

Thanks for the suggestion. We modified and added the sentences as follows:

Therefore, the slightly positive bias of Rayleigh-clear versus GPS-RS winds obtained in this study is almost the same as that obtained by Baars et al. (2020). The bias of Mie-cloudy versus GPS-RS winds is larger than that from Baars et al. (2020). The result that the random error of Mie-cloudy winds is much smaller than that of Rayleigh-clear wind contrasts with our results. The discrepancies are probably caused by different observation location, meteorological conditions, and distance between the measurements.

Line 477 – 479: The sentence is not completely right in my opinion. Martin et al 2021 estimated the representativeness error by considering different error sources, not the representativeness error and error sources. In your case the representativeness errors result from the different measurement geometries and from the collocation criteria (spatial and temporal displacement).

You are right. We corrected the sentence as follows:

Martin et al. (2021) estimated the radiosonde representativeness error $\sigma_{r\_GPS-RS}$ by considering spatial and temporal displacements, and the different measurement geometries of the radiosonde and the Aeolus observations.

Since you use the scaled MAD as estimate for the random error in your paper, I suggest to name the sigma Aeolus 'Aeolus observational error' instead of 'Aeolus random error'. Otherwise, it could be misleading.

Thanks for the suggestion. We changed 'Aeolus observational error' to 'Aeolus random error'.

Summary
Line 502: You say that you investigated the bias dependence on latitude, but there is no such part in the manuscript.

You are right. We removed the term 'latitude'.

Line 521: Again, in my opinion the bias only is getting close to zero towards September 2020. In the early summer months 2020 it is as close to zero than in 2019.

Thanks for the suggestion. We modified the sentence as follows:

The Rayleigh-clear and Mie-cloudy wind biases were close to 0 m s$^{-1}$ towards September 2020.

I miss a nice closing sentence to round off the whole study. You could mention that despite you used three independent data sets with quite different sample sizes the differences in bias estimation are not greater than 1 m/s (except of Rayleigh 2B10). What's your conclusion of the good concordance between the three independent validation data sets? What's the importance of this study for the Aeolus mission and the use of the Aeolus HLOS winds in NWP? You can also give kind of an outlook or suggestions for improvements of your study.

Thanks for the suggestion. We added the sentences in first paragraph of Sect. 6 as follows:

Overall, the systematic errors of the comparisons with the three reference data sets showed consistent tendency. During the baseline 2B02, both Rayleigh-clear and Mie-cloudy winds exhibited positive systematic errors in the ranges of 0.5 to 1.7 m s$^{-1}$ and 1.6 to 2.4 m s$^{-1}$, respectively. The statistical comparisons for the baseline 2B10 period showed smaller biases, –0.8 to 0.5 m s$^{-1}$ for the Ryleigh-clear and –0.7 to 0.2 m s$^{-1}$ for the Mie-cloudy winds. This suggests that the derived systematic errors are due to Aeolus Rayleigh-clear and Mie-cloudy wind systematic errors and not the reference data sets. The reduced bias of the 2B10 period compared to 2B02 is most likely due to the M1 mirror bias correction and the improvement of the hot pixel correction.

And we added the following paragraph in the end of Sect. 6.

To summarize, our validation results obtained from the comparison with the WPRs, CDWLs, and GPS-RSs revealed the quality of the Aeolus Rayleigh-clear and Mie-cloudy HLOS winds over Japan. The systematic errors for the baseline 2B10 were not greater than 1 m s$^{-1}$ and improved as compared with those for the baseline 2B02. The results confirm the necessity to validate the quality of the Aeolus HLOS winds and help to use the Aeolus wind products in NWP data assimilation. Now, we continue to conduct the validation of the Aeolus HLOS winds by using measurements from WPRs and CDWLs. As with this study, the validation activities will provide new insights into the quality of the Aeolus HLOS winds over Japan.

**Technical corrections:**
Line 9: check spelling 'onboard'

We changed 'onboard' to 'on board'

Line 16 - 18: Avoid duplication. Two sentences with the same message. I would suggest the following wording: "The

statistical comparisons for the baseline 2B10 period show smaller biases, –0.8 – 0.5 m s$^{-1}$ for the Ryleigh clear and –0.7 – 0.2 m s$^{-1}$ for the Mie cloudy winds."

Thanks for the suggestion. We changed the sentences as follows:

The statistical comparisons for the baseline 2B10 period showed smaller biases, –0.8 to 0.5 m s$^{-1}$ for the Ryleigh-clear and –0.7 to 0.2 m s$^{-1}$ for the Mie-cloudy winds.

Line 57: please correct 'purposes'

We corrected as suggested.

Line 60: please correct 'requires'

We corrected as suggested.

Line 101 – 106: for better understanding consider to rephrase like this: "In this study, we used three different periods during the processor baseline 2B02 and 2B10 periods to assess L2B data products: 1 October to 18 December 2018 (2B02), 28 June to 31 December 2019 (2B10) and 20 April to 8 October 2020 (2B10). The first period with baseline 2B02 was within the commissioning phase. The L2B data products with the 2B10 baseline include a bias correction for ALADIN's telescope primary (M1) mirror temperature variation (Rennie and Isaksen, 2020; Rennie et al. 2021*) and have been available for new observations since April 2020. The L2B winds from 28 June to 31 December 2019 are a homogeneous reprocessed dataset using also the 2B10 processor version." *I suggest to add the reference Rennie et al. 2021 for more information about the M1 bias correction.

Thanks for the suggestion. We modified the sentences as suggested. We added Weiler et al. (2021b) to the reference.
Weiler, F., Rennie, M., Kanitz, T., Isaksen, L., Checa, E., de Kloe, J., Okunde, N., and Reitebuch, O.: Correction of wind bias for the lidar on-board Aeolus using telescope temperatures, Atmos. Meas. Tech. Discuss. [preprint], https://doi.org/10.5194/amt-2021-171, in review, 2021b.

Line 126: please add a space between number and unit (294 m)

We corrected.

Line 198: it seems that the spacing between the equation environment and the text is missing

We corrected.

Line 205: consider to rephrase as follows: "...estimated HLOS errors are 2.3 m/s during both, baseline 2B01 and 2B10."

Thanks for the suggestion. We modified the phrase as suggested.

Line 228: it seems that the spacing between the equation environment and the text is missing

We corrected.

Line 267 and 292: consider to remove the term 'large' to not sound judgmental

We agreed and removed the term 'large'.

Line 294: almost same sentence than Line 271. I suggest to rephrase as follows: "Again, the discrepancies may be caused by … "

We modified the sentence as suggested.

Line 356: wrong figure reference - Fig. 9d not Fig 9c

We corrected.

Line 440: please modify: "...the reason for that.."

We modified the word as suggested.

Line 440 – 441: consider to rephrase: "Potentially, large horizontal wind gradients in this height region have an influence on the differences.

Thanks for the suggestion. We modified the phrase as suggested.

Line 482: consider to rephrase; "The Aeolus observation error considering the representativeness error in addition to the radiosonde observational error can be calculated as follows:…"

Thanks for the suggestion. We modified the phrase as follows:

The Aeolus random error $\sigma_{Aeolus}$ considering the representativeness error $\sigma_{r\_GPS-RS}$ in addition to the radiosonde observational error can be calculated as follows:

Line 503 and 511: I think there is no need for a paragraph here. Rather, I suggest to put a paragraph before you start to summarize the WPR validation in line 500, like you also did before the CDWL validation and before the radiosonde validation summary. That seems more conclusive for me.

Thanks for the suggestion. We put the second paragraph of Sect. 6 to summarize the WPR validation as suggested.

Line 504 – 505: could remove this sentence. It's kind of repetition of the sentence in line 499 – 500.

We agreed and removed this sentence.

Line 513: be careful with using the expression significant. Actually, this should be based on a statistical significance test. Better remove 'significantly' here.

Thanks for the suggestion. We removed 'significantly'.

Line 518 – 519: this does not make sense. Either "...the bias of Rayleigh-clear HLOS winds were generally negative..." or "...the systematic differences between the Rayleigh clear and WPR HLOS winds were generally negative..."

Thanks for the suggestion. We modified the phrase as suggested.

Line 520: consider to modify: "... and did also not show a clear seasonal trend."

Thanks for the suggestion. We modified the phrase as suggested.

Line 533: next line before the minus symbol

We corrected.

Line 534: again, maybe reconsider the use of significantly here

We removed 'significantly' as suggested.

---

## Author Comment (AC2)

Title: Validation of Aeolus Level 2B wind products using wind profilers, ground-based Doppler wind lidars, and radiosondes in Japan

Author(s): Hironori Iwai et al.

MS No.: amt-2021-243

MS type: Research article

Special Issue: Aeolus data and their application (AMT/ACP/WCD inter-journal SI)

**Responses to referee #2**

First, we would like to thank the referee #2 for your constructive comments that helped us improve our manuscript. In the present document, we provide our responses to the comments. The comments of the referee are reported in black font, our responses and the corresponding modifications in the manuscript in blue font, and the changes to the original manuscript in red font.

**Review Summary**

The paper manuscript "Validation of Aeolus Level 2B wind products using wind profilers, ground-based Doppler wind lidars, and radiosondes in Japan" H. Iwai et al. deals with the validation of Aeolus horizontal line-of-sight (HLOS) winds during different time periods (Oct to Dec 2018, Jun to Dec 2019 and Apr to Oct 2020) based on three different reference data sources coming from windprofilers, two coherent Doppler wind lidars as well as from radiosondes launched from Japan. Hence, these comparisons validate data obtained with the two different Aeolus laser transmitters FM-A and FM-B as well as data based on two different processor versions 2B02 and 2B10. The paper manuscript is clearly structured, all used methodologies are well explained and all figures are clearly visible. The content of the paper is well suited for AMT and especially for the Aeolus special issue. It is recommended to accept the paper manuscript after addressing the minor points that are mentioned below.

We are grateful for the referee's appropriate and positive comments on our manuscript. We revised the manuscript according to your comments and the details are shown as follows.

**- General:**

The entire analysis uses Aeolus data that is processed with two different processor versions 2B02 and 2B10. Could you somehow address the main differences that were implemented in the processor between version 2B02 and 2B10. This would help the reader to understand why one would expect really differences especially in the systematic error due to the implementation of the M1 mirror temperature correction or the hot pixel correction that was implemented.

Thanks for the suggestion. We modified and added the sentences in Sect. 2 as follows:

In this study, we used three different periods during the processor baseline 2B02 and 2B10 periods to assess L2B data products: 1 October 2018 to 15 May 2019 (2B02), 28 June to 31 December 2019 (2B10) and 20 April to 8 October 2020 (2B10). The first period with baseline 2B02 was within the commissioning phase, which was from launch to end of January 2019. The L2B data products with the 2B10 baseline include a bias correction for ALADIN's telescope primary (M1) mirror temperature variation (Rennie and Isaksen, 2020; Weiler et al. 2021b) and have been

available for new observations since April 2020. A hot pixel correction has also been improved in the 2B10 baseline processor version. The L2B winds from 28 June to 31 December 2019 are a homogeneous reprocessed dataset using also the 2B10 processor version. We mainly discuss the measurement performance of Aeolus for Rayleigh-clear and Mie-cloudy winds during the baseline 2B02 and 2B10 periods. The baseline 2B10 period is composed of the M1 mirror and hot pixel bias corrected observations and the reprocessed data set.

We added the sentence in Sect. 5.1.1 as follows:

The reduced bias of the baseline 2B10 period compared to the baseline 2B02 is most likely due to the M1 mirror bias correction (Rennie and Isaksen, 2020; Weiler et al. 2021b) and the improvement of the hot pixel correction.

We added Weiler et al. (2021b) to the reference.

Weiler, F., Rennie, M., Kanitz, T., Isaksen, L., Checa, E., de Kloe, J., Okunde, N., and Reitebuch, O.: Correction of wind bias for the lidar on-board Aeolus using telescope temperatures, Atmos. Meas. Tech. Discuss. [preprint], https://doi.org/10.5194/amt-2021-171, in review, 2021b.

You compare your determined results with different studies (GPS-RS results with Martin et al.; CDWL results with Witschas et al., WPR results with Guo et al.). But actually, you can use the data of all three instruments as reference. Thus, I am missing a detailed comparison of your own results from the different measurement data of the WPR, the CDWL and the GPS-RS. This should also be available in the summary, maybe with a table that summarizes to retrieved systematic errors based on the respective reference data sets.

Thanks for the suggestion. We added the sentence in first paragraph of Sect. 6 as follows:

Overall, the systematic errors of the comparisons with the three reference data sets showed consistent tendency. During the baseline 2B02, both Rayleigh-clear and Mie-cloudy winds exhibited positive systematic errors in the ranges of 0.5 to 1.7 m s$^{-1}$ and 1.6 to 2.4 m s$^{-1}$, respectively. The statistical comparisons for the baseline 2B10 period showed smaller biases, –0.8 to 0.5 m s$^{-1}$ for the Ryleigh-clear and –0.7 to 0.2 m s$^{-1}$ for the Mie-cloudy winds. This suggests that the derived systematic errors are due to Aeolus Rayleigh-clear and Mie-cloudy wind systematic errors and not the reference data sets. The reduced bias of the 2B10 period compared to 2B02 is most likely due to the M1 mirror bias correction and the improvement of the hot pixel correction.

Sometimes there is a confusion with the "-" sign which sometimes denotes a minus and sometimes denotes a "to". Thus, it is suggested to replace the "-" sign with the word "to" and only use "-" as a minus sign.

Thanks for the suggestion. We replaced the "−" sign with the word "to" and only use "−" as a minus sign.

If possible, I would not spread units over two lines.

Since we don't know where the line break positions will be after the final proofreading, we think that this work is meaningless.

- **Fig.1, caption:** bule → blue

We corrected.

**- Line 92:** "two interferometers"…actually, as you also write later in the text, there are three interferometers: one Fizeau interferometer, and two Fabry-Perot interferometers all of them illuminated sequentially.

We changed "two interferometers" to "three interferometers".

**- Line 98:** "Fizeau and Fabry Perot as narrowband filters" Somehow this is true, but the bandwidth is still different by a factor of 5 (Fizeau interferometer about 2 GHz, Fabry-Perot interferometers about 11 GHz).

We removed this sentence.

**- Line 99:** "After passing through some optics". You could skip that part of the sentence as it provides no information.

Thanks for the suggestion. We modified the phrase as follows:

after passing through relay optics

**- Line 106:** "Commissioning phase" which was from x.x to x.x?.

We added the phrase as follows:

commissioning phase, which was from the launch of Aeolus to the end of January 2019

**- Line 110:** "Several different technical, instrumental, and retrieving checks account for this flag." → Could you give a few examples which checks are performed and considered for deriving the validity flag?

We added the phrase as follows:

for example checking for signal and background radiation levels

**- Line 126:** "There is no significant difference between wind profiler winds and radiosonde winds in the biases and root mean square errors. → What do you mean with "significant"? Is there a difference? If yes, it would be better to quantify here. How do you determine the bias of WPR measurements or rather radiosondes?

We removed this sentence and added the sentences and a reference as follows:

The wind measurement accuracy of the WPRs was evaluated by comparisons with winds forecasted by the NWP model and radiosondes (Tada 2001). From the comparisons, the wind measurement accuracy of the WPRs was comparable to that from radiosonde observations.

Tada, H.: Use of wind profiler data in Japan. Technical Report on Numerical Weather Prediction (in Japanease), 34, 55–58, Numerical Prediction Division, Japan Meteorological Agency, 2001.

**- Line 126:** "294m" → 294 m (space)

We corrected.

**- Line 148:** "Doppler beam swinging (DBS) technique ". Is this a well-known technique? Could you add a reference here?

Thanks for the suggestion. We added Röttger and Larsen (1990) to the reference.

**- Line 149:** "The Doppler velocity spectra for all range bins of each beam were obtained 10,000 times on average. Since the PRF was 10 kHz, the accumulation time of each beam was 10 s". If I understand it correctly, the averaging time is one second, or you accumulate 100000 spectra. Or do I misunderstand something here?

You are right. We replaced '10,000' to '100,000'.

**- Line 152:** "Maximum likelihood estimator" Can you give a reference here?

Thanks for the suggestion. We added Levin (1965) to the reference.

Levin, M. J.: Power spectrum parameter estimation, IEEE Trans. Inform. Theory, 11, 100–107, https://doi.org/10.1109/TIT.1965.1053714, 1965.

**- Line 153:** "Bias was estimated at 0.02 m s$^{-1}$ using measurements from a stationary hard target." → I guess, this value is only true for single LOS measurements, isn't it? If yes, this should be clarified here.

Thanks for the suggestion. We added the phrase "for single LOS measurements".

**- Line 159:** "As mentioned earlier, we averaged Doppler velocity spectra for all range bins of each beam from 30 min before to 30 min after the passage of Aeolus, and then the vertical profiles of horizontal wind speed and wind direction were acquired by the DBS technique." → Is it true that you averaged all spectra for 60 minutes and then you determine the mean wind speed? Using such an approach, I would expect the center peak in your power spectrum to be rather broad. Wouldn't it be better to calculate the wind speed on a e.g. one-minute average, and then calculate the mean over the 60 data points? Furthermore, I do not understand why you calculate and average for your CDWL comparison. Couldn't you actually just use the profile measured directly during the Aeolus overpass?

You are right. We averaged all spectra for 60 minutes and then estimated the LOS wind speeds of each beam.

We compared the Doppler velocity widths estimated from 60-min-averaged Doppler velocity spectra ($\sigma_{60min}$) with those estimated from 10-s-averaged Doppler velocity spectra during the Aeolus overpass ($\sigma_{10s}$). At Kobe, $\sigma_{10s}$ = 0.51±0.18 m s$^{-1}$ and $\sigma_{60min}$ = 0.78±0.31 m s$^{-1}$. At Okinawa, $\sigma_{10s}$ = 0.59±0.18 m s$^{-1}$ and $\sigma_{60min}$ = 0.79±0.19 m s$^{-1}$. We calculated the mean differences (BIAS) and the standard deviation (STD) of the difference between $\sigma_{10s}$ and $\sigma_{60min}$. The BIAS is 0.26 (0.19) m s$^{-1}$ and the STD is 0.27 (0.20) m s$^{-1}$ at Kobe (Okinawa). Since the Doppler velocity resolution of the CDWLs is 0.75 m s$^{-1}$, the longer spectral averaging time (60 min) have a small impact on the Doppler velocity estimate.

The horizontal resolutions of the Aeolus wind observations are about 90 km for the Rayleigh channel and 90 (or 10) km for the Mie channel, so there is no sense in comparing instantaneous measurements obtained from the CDWLs during the Aeolus overpass. Although the longer spectral averaging time (60 min) yield broader power spectra, that is needed for the higher altitude where the signal-to-noise ratio (SNR) is much lower. Even if it is difficult to detect the spectral peak and estimate the Doppler velocity due to the low SNR for the accumulation time of 10 s, the probability of the peak detection and Doppler velocity estimation gets higher for the longer spectral averaging time.

**- Line 229:** "Note that this estimation of $\sigma$ includes the representativeness error due to the spatial and temporal mismatch between Aeolus and reference instruments' measurements."

→ The representativeness error is likely to be different for the different reference instrument measurements. For instance, for the CDWL measurements you could decrease the representativeness error by decreasing the temporal averaging time to only a few minutes. Have you tried if this changes the random error for the comparison?

If the temporal averaging time decreases to only a few minutes, the numbers of data pairs for the comparison decrease and the derived random errors are not reliable.

**- Line 248:** "Furthermore, the range-gate settings of Aeolus were changed on 26 February 2019, which also increased the number of available data points during the baseline 2B10 period." → What was changed in particular? More range gates in the troposphere, less in the stratosphere? Or just increasing the range bin size? Here it would be helpful to get more details.

Thanks for the suggestion. We modified this sentence as follows:

The range-bin settings of Aeolus were changed on several occasions (Rennie and Isaksen, 2020). The number and resolution of the bins in the lower troposphere increased after 21 October 2019. Therefore, the number of available Rayleigh-clear and Mie-cloudy winds for the comparison increased during the baseline 2B10 period.

**- Line 252:** "slightly positive". As the bias is 1.6 to 1.8 m/s which is a factor of more than two larger the originally specified, I would skip the word "slightly" here.

We removed the word "slightly" as suggested.

**- Line 261:** "Comparison to A2D data". → You should be careful when comparing to A2D data as they compare LOS winds and not HLOS winds.

Thanks for the suggestion. We modified the sentences in the second paragraph of Sect. 5.1.1 as follows:

Lux et al. (2020b) compared the Rayleigh-clear winds measured along the Aeolus LOS with LOS winds measured with the ALADIN Airborne Demonstrator (A2D) during the WindVal III validation campaign carried out in central Europe from 17 November to 5 December 2018 (i.e., during the baseline 2B02 period). They reported a bias of 2.56 m s$^{-1}$ with a scaled MAD of 3.57 m s$^{-1}$, corresponding to HLOS values of 4.25 and 5.93 m s$^{-1}$, respectively.

And we removed the sentence "They also reported that the slope of the linear regression line and the correlation coefficient of Rayleigh-clear versus A2D winds were 0.83 and 0.80, respectively."

**- Line 289:** It is worth mentioning here that only ascending orbits were underflown during the WindVal III campaign.

Thanks for the suggestion. We added the following sentence:

It is note that the WindVal III flights were conducted for probing the ascending orbit.

**- Line 311:** "The main reason for not yet achieving the mission requirement for random errors is probably related to the large representativeness error due to the large sampling volume of the WPR." → Also the Aeolus laser pulse energy is remarkably less than specified. With the representativeness error you would argue that the actual random error of Aeolus L2B data could meet the requirements. However, this is not true.

Thanks for the suggestion. We modified the sentence as follows:

The main reason for not yet achieving the mission requirement for random errors is the lower laser energy compared to the anticipated 80 mJ (Reitebuch et al. 2020a and 2020b). Additionally, the large representativeness error due to the large sampling volume of the WPR is probably related to the larger Aeolus random error.

We modified the related sentence in Abstract as follows:

The main reason for the large Aeolus random errors is the lower laser energy compared to the anticipated 80 mJ. Additionally, the large representativeness error of the WPR is probably related to the larger Aeolus random error.

We modified the related sentences in Sect. 6 as follows:

The main reason for the large Aeolus random errors is the lower laser energy compared to the target of 80 mJ. Additionally, the large representativeness error due to the large sampling volume of the WPR is probably related to the larger Aeolus random error.

- **Line 320:** "…there are relatively many paired data points for comparison (Fig. 6a)." → What does relatively many mean? Can you please quantify. It would also be very helpful to have this information plotted in Fig.6/Fig.7. This would give the possibility to understand how likely the shown bias trend is. For instance, is the negative Mie bias for descending orbits in 10 km altitude real, or just a result of very few data points and thus not reliable.

Thanks for the suggestion. Since it is difficult to plot the number of compared data points in Figs. 6 and 7, we plotted the vertical profile of the number of compared data points as shown in Figs. S1 and S2. We removed the phrase "where there are relatively many paired data points for comparison". We added the sentence as follows:

[Figure]

**Figure S1.** Vertical profiles in 1 km bins of the number of compared data points between the Aeolus and WPR HLOS winds for (a, b, c) Rayleigh-clear winds and (d, e, f) Mie-cloudy winds for (a, d) all data and (b, e) ascending and (c, f) descending orbits for baseline 2B02.

But there are very few paired data points in 10 km altitude (Figs. S1e and S1f) and thus the biases in 10 km altitude are not reliable.

[Figure]

Figure S2. Same as Fig. S1 but for baseline 2B10.

- **Line328:** "The systematic error was less than that of the baseline 2B02." → Why? Here, you could refer to the differences in 2B02 and 2B10 which I suggested to discuss in the previous part of the manuscript.

Thanks for the suggestion. We modified the sentence as follows:

The systematic error is less than that of the baseline 2B02 due to the M1 mirror bias correction (Rennie and Isaksen, 2020; Weiler et al. 2021b) and the improvement of the hot pixel correction (see Sect. 5.1.1).

- **Line 332:** "However, this result is different from that in the other validation studies conducted during the baseline 2B10 period (Guo et al., 2021)." → What is different? What is the result by Guo et al.? Would be good to write one sentence here such that this information is available without reading Guo et al., 2021.

Thanks for the suggestion. We added the sentence as follows:

Guo et al. (2021) reported a large bias of 3.23 m s$^{-1}$ with a standard deviation of 17 m s$^{-1}$ for the Rayleigh-clear winds in the altitude range of 0 to 1 km.

- **Line 339:** "(descending) orbit, the minimum (maximum) bias is –1.93 (0.54) m s–1 in the altitude range of 5–6 (4–5) km." → this is confusing. Is the bias of -1.93 m/s corresponding to ascending or descending orbits?

We removed this sentence because this is not true.

**- Fig. 8:** "Monthly averages" → Why do you calculate monthly averages and do not show a time series on a daily basis?

Since there are only a few data pairs per day, the results from daily averages are statistically unreliable. Therefore, monthly averages are needed to obtain statistically meaningful analysis results.

**- Line 358:** "because the Mie return signal does not depend on the laser energy (Martin et al., 2021)." → I would not write it that strong. Indeed, compared to Rayleigh returns, Mie signals are much more depending on the atmospheric backscatter. But of course, if the laser pulse energy is too low, one would not be able to measure at all.

You are right. We corrected the sentence as follows:

There is no significant increase in the standard deviations of Mie-cloudy winds with time, because the Mie return signal does not only depend on the laser energy but also on the presence of aerosols or clouds (Martin et al., 2021).

**- Line 369:** "During the baseline 2B02 period, the bias of Rayleigh-clear and WPR HLOS winds slightly increased as the scattering ratio increased (Fig. 10a)." → Would this be better visible when calculating daily means instead of monthly means?

The results shown in Fig. 10 were calculated using all data during the baseline 2B02 and 2B10 periods. Since there are only a few data pairs per day, the results obtained from daily averages are statistically unreliable.

**- Line 391:** "result is similar to that in the comparisons of Aeolus and WPR measurements." → which provides biases of x.x m/s. Would be good to repeat the numbers here.

Thanks for the suggestion. The following phrase was added:

which provides biases of 1.69 m s$^{-1}$ (Rayleigh) and 2.42 m s$^{-1}$ (Mie)

**- Line 394:** "The values are smaller than the scaled MADs of Rayleigh-clear (Mie-cloudy) versus WPR winds." → Why do you think is this the case? The representativeness is similar, and the respective measurements errors of the WPR or rather the CDWL is corrected, isn't it?

Thanks for the suggestion. We added the sentence as follows:

The main reason for the difference is probably related to that the random error is larger for the WPR (3 m s$^{-1}$) than for the CDWL (2 m s$^{-1}$).

**- Line 423:** "The Rayleigh-clear winds show good coverage and closely follow the shape of the wind profile at altitudes higher than 2 km." → Any ideas or hints what causes outliers as e.g. the one at 8 km for the Rayleigh-clear winds? Still aerosol contamination, as the range gates below provides a valid Mie wind? Would be interesting to see if there is an issue with the cross-talk correction of Mie signals in the Rayleigh channel...

Thanks for the suggestion. There are large deviations between Rayleigh-clear and GPS-RS winds at 3 km and 8 km. The scattering ratio is 1.15 and the relative humidity is about 30% at 3 km. Although there is a valid Mie wind at 3 km, the Mie wind is filtered out due to the HLOS error threshold of 5 m s$^{-1}$. This suggests that there are issues with the atmospheric classification and the cross talk correction of Mie signals in the Rayleigh channel. At 8 km there is

no valid Mie wind. The scattering ratio on the Rayleigh channel is 1.13 and the relative humidity obtained from the GPS-RS is about 20% at 8 km. This suggests that there was a small influence of the cross talk of Mie signals to the Rayleigh channel on the large deviation at 8 km. We modified and added the sentences as follows:

The Rayleigh-clear winds show good coverage and closely follow the shape of the wind profile at altitudes higher than 2 km, but there are large deviations between Rayleigh-clear and GPS-RS winds at 3 km and 8 km. The scattering ratio on the Rayleigh channel is 1.15 and the relative humidity obtained from the GPS-RS is about 30% at 3 km. Although there is a valid Mie-cloudy wind at 3 km, it is filtered out due to the HLOS error threshold of 5 m s$^{-1}$. This suggests that the atmospheric classification in the Rayleigh channel was not working properly and the cross talk of Mie signals in the Rayleigh channel could have led to the large deviation. At 8 km, there is no valid Mie-cloudy wind. The scattering ratio and relative humidity are 1.13 and about 20%, respectively. This suggests that the cross talk has a small influence on the large deviation. The reason for that is unclear. Since the horizontal distance between the Rayleigh-clear measurements and the GPS-RS is about 80 km in this height region, large horizontal wind gradients in this height region potentially have an influence on the deviation.

**- Line 435:** "…the clouds were partly existent in the Aeolus observational domain" → This means that clouds are not sufficiently corrected or filtered out in the Rayleigh data product? E.g., the Rayleigh-clear wind in between 3-4 km where also clouds were partly present shows a quite large bias. Is there any way to analyze this altitude in more detail, e.g., by analyzing single Aeolus "measurements" instead of "observations", or is this information not contained in the L2B data product?

Thanks for the suggestion. Since the relative humidity obtained from the GPS-RS was about 90% at 3 to 4 km altitude, this supports that the clouds were existent in the altitude range. The Rayleigh-clear wind shows a large bias at 3 to 4 km altitude, but the scattering ratio on the Rayleigh channel was 1.15. This suggests that there was an issue with the cross talk correction of Mie signals in the Rayleigh channel. We modified and added the sentences as follow:

The occurrence of cloud was sporadically detected by the CDWL and the relative humidity obtained from the GPS-RS was about 90% at 3 to 4 km altitude (not shown). It is assumed that the clouds were partly existent in the Aeolus observational domain. The Rayleigh-clear wind shows a large bias at 3 to 4 km altitude, but the scattering ratio on the Rayleigh channel was 1.15. This suggests that there was an issue with the cross talk correction of Mie signals in the Rayleigh channel.

**- Line 440:** "There is a possibility regarding horizontal wind gradients in this height region." → You have actual wind speed and direction available from the radiosonde, right? If yes, it might be good to show if there was indeed a wind shear in this altitude.

Thanks for the suggestion. The vertical shear of the GPS-RS HLOS wind is about 10 m s$^{-1}$ km$^{-1}$ in the altitude range of 11 to 12 km (Fig. 13b). Since the range-bin resolution in this altitude range is 1 km, it is expected that the difference between the Rayleigh-clear HLOS wind and GPS-RS HLOS wind is negative. Therefore, the positive difference is potentially influenced by large horizontal wind gradients in this altitude range. We modified the sentence as follows:

Potentially, large horizontal wind gradients in this height region have an influence on the differences.

**- Line 477:** "Martin et al. (2021) estimated the radiosonde representativeness error, and error sources caused by spatial and temporal displacements need to be considered," → How do they do that? Based on comparison to measurements or with respect to theoretical assumptions? This would be an interesting side note.

Thanks for the suggestion. We modified this sentence as follows:

Martin et al. (2021) estimated the radiosonde representativeness error $\sigma_{r\_GPS-RS}$ by considering spatial and temporal displacements, and the different measurement geometries of the radiosonde and the Aeolus observations.